# The Complexity of Finding Local Optima in Contrastive Learning

Jingming Yan[*1], Yiyuan Luo[*2], Vaggos Chatziafratis[2,3], Ioannis Panageas[1,3],
Parnian Shahkar[1], and Stelios Stavroulakis[1]

[1]University of California, Irvine
[2]University of California, Santa Cruz
[3]Archimedes AI

## Abstract

Contrastive learning is a powerful technique for discovering meaningful data representations by optimizing objectives based on *contrastive information*, often given as a set of weighted triplets $\{(x_i, y_i^+, z_i^-)\}_{i=1}^m$ indicating that an "anchor" $x_i$ is more similar to a "positive" example $y_i$ than to a "negative" example $z_i$. The goal is to find representations (e.g., embeddings in $\mathbb{R}^d$ or a tree metric) where anchors are placed closer to positive than to negative examples. While finding *global* optima of contrastive objectives is NP-hard, the complexity of finding *local* optima—representations that do not improve by local search algorithms such as gradient-based methods—remains open. Our work settles the complexity of finding local optima in various contrastive learning problems by proving PLS-hardness in discrete settings (e.g., maximize satisfied triplets) and CLS-hardness in continuous settings (e.g., minimize Triplet Loss), where PLS (Polynomial Local Search) and CLS (Continuous Local Search) are well-studied complexity classes capturing local search dynamics in discrete and continuous optimization, respectively. Our results imply that no polynomial time algorithm (local search or otherwise) can find a local optimum for various contrastive learning problems, unless PLS $\subseteq$ P (or CLS $\subseteq$ P for continuous problems). Even in the unlikely scenario that PLS $\subseteq$ P (or CLS $\subseteq$ P), our reductions imply that there exist instances where local search algorithms need exponential time to reach a local optimum, even for $d = 1$ (embeddings on a line).

## 1  Introduction

Extracting meaningful representations from complicated datasets is a cornerstone of machine learning. For the past decades, algorithmic questions of how to find convenient representations (Euclidean space, tree metrics, etc.) faithfully capturing distance relationships have been at the forefront of metric embeddings and multidimensional scaling [Kruskal, 1964a, Borg and Groenen, 2007, Indyk et al., 2017], yielding by now a vast literature with both practical successes and deep mathematical insights.

Due to the high cost of labeling datasets and obtaining accurate distances, many communities focus instead on learning representations based on easier-to-obtain *contrastive information*, i.e., distance comparisons [Agarwal et al., 2007, Tamuz et al., 2011, Jamieson and Nowak, 2011, Van Der Maaten and Weinberger, 2012, Terada and Luxburg, 2014, Jain et al., 2016, Kleindessner and von Luxburg, 2017]. Contrastive information, like "item $x$ is closer to item $y$ than to $z$" or "among $x, y, z$, items $x, z$ are farthest apart" is much easier to obtain than numerical values ("how similar is $x$ to $y$"), essentially creating pseudo-labels with little to no supervision. Indeed, such triplets are standard in contrastive learning, e.g., the popular "anchor-positive-negative" paradigm $(x, y^+, z^-)$ [Schroff et al., 2015],

---

[*]Equal contribution. Corresponding authors jingmy1@uci.edu, yluo124@ucsc.edu.

39th Conference on Neural Information Processing Systems (NeurIPS 2025).

since image transformations (cropping, rotations) or nearby words produce anchor-positive pairs, and random images/words yield anchor-negative pairs (see also "hard negatives" [Robinson et al., 2020]).

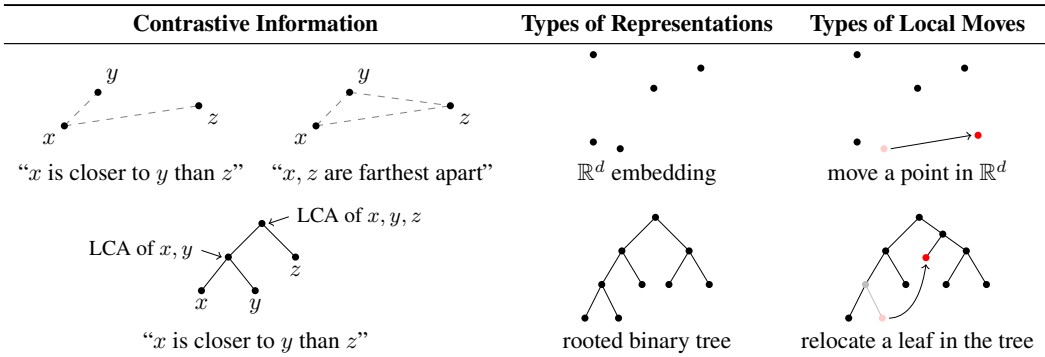

Figure 1: An overview of contrastive learning problems studied here (LCA: lowest common ancestor).

Interestingly, even though contrastive information lies at the heart of contrastive learning pipelines, the history of using ordinal information, i.e., comparisons instead of absolute numerical values, dates back to the 1960s in psychometrics and (non-metric) multi-dimensional scaling [Torgerson, 1952, Thurstone, 1954, Kruskal, 1964b]. Since then, *ordinal* embeddings (also monotone or contrastive embeddings [Bilu and Linial, 2005, Avdiukhin et al., 2024, Chen et al., 2022, Chatziafratis and Indyk, 2024]) are widely-studied in computer science, because important applications (nearest-neighbor, recommendation, ranking, crowdsourcing) only care to preserve the relative ordering of distances (not lengths) [Agarwal et al., 2007, Bădoiu et al., 2008, Tamuz et al., 2011, Vankadara et al., 2019, Ghosh et al., 2019].

Despite empirical successes, various aspects of contrastive learning are not well-understood. To address this, works on theoretical foundations have focused on generalization [Alon et al., 2023], inductive bias [Saunshi et al., 2022, HaoChen and Ma, 2022], latent classes [Saunshi et al., 2019], hard negatives [Robinson et al., 2020, Kalantidis et al., 2020, Awasthi et al., 2022], multi-view redundancy [Tosh et al., 2021], mutual information [Oord et al., 2018, Hjelm et al., 2018] and more.

**Optimization of contrastive objectives.** Our paper focuses on *optimization* aspects of contrastive learning with widely-used objectives, both in discrete and continuous settings (Figure 1). The input is a set $\mathcal{C}$ of $m$ triplets $\{(x_i, y_i^+, z_i^-)\}_{i=1}^m$ on a dataset $\mathcal{S}$, with non-negative weights $w_i \geq 0$ indicating the importance of each constraint (see [Robinson et al., 2020, Kalantidis et al., 2020] for benefits of "hard negatives" which are difficult to distinguish from anchor points). The task is to find a representation $f(\cdot)$ respecting the given constraints as much as possible by optimizing Objectives 1, 2, 3 below:

1. Triplet Maximization in $\mathbb{R}^d$: Many algorithmic works in contrastive embeddings, aim at maximizing the weight of *satisfied* triplets. We say that a triplet $(x_i, y_i^+, z_i^-)$ is satisfied by the embedding $f(\cdot)$, if $\|f(x_i) - f(y_i)\|_2 \leq \|f(x_i) - f(z_i)\|_2$. Even the case of $d = 1$ (line embedding or ranking) is well-motivated and non-trivial [Arora et al., 1995, 2002, Guruswami et al., 2011, Fan et al., 2020].

2. Triplet Maximization on trees: Here, we map data onto leaves of a tree $T$ maximizing the weight of satisfied triplets ($\text{dist}_T(x_i, y_i) \leq \text{dist}_T(x_i, z_i)$, or equivalently, $T$ has a subtree containing $x_i, y_i$, but not $z_i$). Such hierarchical clustering problems known as triplet reconstruction or consistency naturally appear across areas [Aho et al., 1981, Byrka et al., 2010, Vikram and Dasgupta, 2016, Bodirsky et al., 2017, Chatziafratis et al., 2018, Emamjomeh-Zadeh and Kempe, 2018].

3. Minimize Triplet Loss in $\mathbb{R}^d$: The influential FaceNet paper [Schroff et al., 2015] introduced the Triplet Loss, which has since evolved into one of the most prominent contrastive losses:

$$\mathcal{L}(\mathcal{C}, f(\cdot)) := \sum_{i=1}^m w_i \mathcal{L}_i, \text{ where } \mathcal{L}_i := \max\{\|f(x_i) - f(y_i)\|_2^2 - \|f(x_i) - f(z_i)\|_2^2 + \alpha, 0\}.$$

The margin $\alpha$ specifies the minimum gap of distance $(x_i, y_i)$ and $(x_i, z_i)$ and $w_i$ is the importance of a triplet. This loss "pushes" positive pairs close together while keeping negative pairs far apart.

Our primary motivation is to understand the complexity of finding representations based on contrastive information. Unfortunately, NP-hardness results prevent us from finding *globally* optimum representations in the worst-case, and so in practice, local search algorithms are deployed. Local search is a general algorithmic approach that examines improving moves from a set of allowable nearby configurations to the current solution (most notably gradient-based methods and various heuristics in discrete optimization [Orlin et al., 2004]). We ask the following basic questions:

> **Motivating Questions:** *How hard is it to find a* locally optimum *representation for contrastive learning? Can we find a local optimum in polynomial time? How efficient is local search?*

In this context, a locally optimum representation is one that cannot be improved by taking further steps of gradient-based methods, or generally *any* type of local search algorithm.

## 1.1 Our contributions

Our main contribution is to formally study the above questions from the lens of computational complexity, and give strong evidence that for many problems with contrastive objectives, even finding a *locally* optimum solution for the objectives is intractable in a specific formal sense. Local search is a widely-used heuristic for many NP-hard problems [Lawler, 1985, Schäffer and Yannakakis, 1991, Ahuja et al., 2002, Orlin et al., 2004], wherein we iteratively perform local moves (e.g., gradient step or reassignment of points) that get better objective value, terminating at a solution with no improving move, i.e., a *local* optimum. Of course, a local optimum need not be a global optimum, and since there are more local optima than global optima in general, local optima may appear easier to find. Nevertheless, there is currently an overall lack of theoretical guarantees regarding local optimization of the contrastive objectives 1, 2, 3; indeed, it is not even clear whether there is a polynomial-time algorithm for computing a local optimum. Our main result is to show that no such polynomial-time algorithm exists (unless there is an unlikely collapse of complexity classes, see below). Moreover, an unconditional consequence of our reductions, is that there exist contrastive learning instances where local search algorithms (including gradient-based methods) require *exponential* time before reaching a local optimum. To the best of our knowledge, we are the first to formally investigate the complexity of local solutions in contrastive learning objectives.

**Proving hardness of local optima.** The remarkable theory of NP-hardness was developed to understand the difficulty of computing the value of a *global* optimum in optimization problems. Hence, NP-hardness is not suitable to describe local optimality, which corresponds to fixed points of local search algorithms. Johnson, Papadimitriou, and Yannakakis [1988] initiated the complexity-theoretic study of discrete local search problems by defining the complexity class PLS (Polynomial Local Search), and later, Daskalakis and Papadimitriou [2011] defined the class CLS (Continuous Local Search) for continuous local search problems. Due to the significance of computing fixed points, both PLS and CLS have played a prominent role in optimization and algorithmic game theory. For example, computing a pure Nash equilibrium in congestion games is PLS-hard [Fabrikant et al., 2004], and it was recently shown that computing a Karush-Kuhn-Tucker point (fixed points of gradient-descent) of a quadratic program is CLS-hard [Fearnley et al., 2024]. Surprisingly, even though PLS and CLS were originally defined with very different problems in mind, a recent breakthrough [Fearnley, Goldberg, Hollender, and Savani, 2023] established a deep connection between them,* i.e., that CLS = PLS ∩ PPAD. Building on it, Babichenko and Rubinstein [2021], Anagnostides et al. [2023], Ghosh and Hollender [2024], Anagnostides et al. [2025] have shown that broad classes of games, including congestion games and adversarial team games, are captured by CLS. Beyond games, for connections of PLS and CLS to cryptography, see Bitansky and Gerichter [2020], Hubácek and Yogev [2020]. Our work provides formal evidence of computational intractability of finding local optima in contrastive learning:

**Theorem** (Abridged; see Theorems 3.2, 3.3, 3.4, 4.1). *Triplet maximization problems 1, 2 are* PLS-*hard, i.e., every problem in* PLS *efficiently reduces to them. The Triplet Loss minimization problem 3 is* CLS-*hard, i.e., every problem in* CLS *efficiently reduces to it. As a corollary, assuming the widely-believed* PLS ⊄ P *and* CLS ⊄ P*, no polynomial time algorithm (local search or otherwise) can find a local optimum of objectives 1, 2, 3. Even if* PLS ⊆ P *or* CLS ⊆ P*, there exist instances where local search (including gradient-methods) require exponential time to reach a local optimum.*

---

*PPAD captures computation of mixed Nash equilibria, see celebrated work of Daskalakis et al. [2009].

Formal statements are in Section 3, 4. As is common in complexity, polynomial time refers to algorithms whose runtime is polynomial in the input size (provided in binary representation). Our results also extend to *approximate* local optima (solutions within a small $\epsilon > 0$ from a local optimum).

**Challenges and proof ideas.**    The consequence of proving that a problem is PLS-hard (or CLS-hard) is that in a certain precise sense, such problems are as hard as any other local search problem where the goal is to find a local optimum. Here, local optimality is defined with respect to a generic definition of a local search algorithm [Johnson et al., 1988, Daskalakis and Papadimitriou, 2011]. The main technical challenge in all works establishing hardness results is how to do PLS-reductions (or CLS-reductions). Roughly speaking, these are special type of efficient transformations between problems, that preserve *local* optimality: the intuition is that starting from an already-known hard problem, and using a PLS-reduction to another problem, e.g., the contrastive objectives above, then any efficient algorithm that purportedly finds a local optimum of the latter, would in fact compute a local optimum of the original (known-to-be-hard) problem. Contrast this with the common NP-reductions that are only required to preserve the value of global optima, ignoring how local solutions are being altered.

Our results provide PLS-reductions from the LocalMaxCut problem [Schäffer and Yannakakis, 1991] (see Section 2 for definitions) to the maximization of satisfied triplets (Obj. 1, 2), and a CLS-reduction from QuadraticProgram-KKT [Fearnley et al., 2024] to the Triplet Loss (Obj. 3). Our PLS-reductions rely on novel gadgets that allow us to encode graph cuts and local (vertex) moves via triplet constraints and embeddings in Euclidean space, trees or even the line (for the case of rankings). Our gadgets impose a series of "heavy" contrastive triplet constraints on a special set of "boundary" points, thus ensuring they are not allowed to move back and forth, otherwise the objective value would drift away from any local optimum. After an embedding is performed, the two sides of the graph cut can be formed by looking at which nodes were placed to the "left" or "right" of the boundary points. Regarding our CLS-reductions, we encode and recover KKT stationary points of quadratic programs $\min_{\boldsymbol{x} \in [0,1]^n} \boldsymbol{x}^\top \mathbf{Q} \boldsymbol{x} + \boldsymbol{b}^\top \boldsymbol{x}$, by finding local minima of Triplet Loss, even if the representation $f(\cdot)$ comes from a simple linear embedding model parameterized by $\boldsymbol{\theta} \in [0,1]^n$ such that $f_{\boldsymbol{\theta}}(\boldsymbol{x}) = \boldsymbol{\theta}^\top \boldsymbol{x}$. Towards this, we first break the quadratic form into triples of variables $x, y, z \in [0, 1]$ and we generate a specially crafted collection $\mathcal{T}$ of contrastive triplets $(x_i, x_j^+, x_k^-)$, where $x_i, x_j, x_k$ are coordinates of $\boldsymbol{x}$. However, contrastive constraints introduce dependencies among their shared variables and to deal with the interacting terms, we introduce groups of contrastive triplets with carefully chosen weights that depend on the coefficients of the quadratic form.

## 2    Preliminaries

### 2.1    Discrete objectives in contrastive learning and PLS

Johnson et al. [1988] defined PLS to describe local search problems. A problem $\Pi$ is in PLS if the following three polynomial-time algorithms exist:[†] (i) the first algorithm, given an instance of $\Pi$, it outputs an arbitrary feasible solution $S$, (ii) the second, given $S$, it returns a number which is the objective value of the feasible solution, and (iii) the third, given $S$, it either reports "locally optimal" or produces a better solution. Implicit in the definition is the fact that feasible solutions have polynomially many *neighboring* solutions (the third algorithm is polynomial-time). In this sense, one can think of PLS as problems with efficient verification of *local* optimality.

**Definition 2.1** (PLS-reduction). A PLS-reduction from problem $\Pi_1$ to problem $\Pi_2$ is two polynomial-time algorithms: (i) the first algorithm $A$ maps every instance $x$ of $\Pi_1$ to an instance $A(x)$ of $\Pi_2$, and (ii) the second algorithm $B$ maps every local optimum of $A(x)$ to a local optimum of $x$.

A PLS-reduction ensures that if we find a local optimum for $\Pi_2$ in polynomial time then, we could also find a local optimum for $\Pi_1$ in polynomial time. A problem is PLS-hard if every problem in PLS can be reduced to it. In complexity, it is widely-believed that PLS $\not\subseteq$ P, hence there is no polynomial-time algorithm for computing a local optimum of a PLS-hard problem. Interestingly, Schäffer and Yannakakis [1991] proved many natural problems are PLS-hard, including LocalMaxCut:

**LocalMaxCut Problem.**
Input : A weighted undirected graph $G(V, E)$ with a non-negative weight $w_e \geq 0$ for each edge.

---

[†]Both PLS and CLS can be equivalently defined with arithmetic circuits [Fearnley et al., 2023]

> OUTPUT : A partition $(S, \overline{S})$ of the vertices $V$ in two nonempty sets, such that no vertex $v$ can increase the value of the cut, i.e., the sum of weighted edges cut, by switching sides.

We now define widely-used (discrete) objectives of maximizing satisfied contrastive constraints. To simplify notation, from now on we drop the signs and simply write $(x_i, y_i, z_i)$ for contrastive triplets. All embeddings here map a set of $n$ items to non-overlapping points of a metric space of dimension $d \leq n$.

### LOCALCONTRASTIVE-EUCLIDEAN Problem.

INPUT : Set $V$ of $n$ vertices, together with $m$ contrastive triplets $\{(x_i, y_i, z_i)\}_{i=1}^{m}$ where $x_i, y_i, z_i \in V$ (we dropped the $+/-$ signs for lighter notation), and target dimension $d$. Each triplet has a non-negative weight $w_i \geq 0$.

OUTPUT : An embedding $f : V \to \mathbb{R}^d$ such that[a] no vertex $v$ can increase the value of the embedding by switching its location in $\mathbb{R}^d$. We say a triplet $(x_i, y_i, z_i)$ is *satisfied* by $f(\cdot)$, if $x_i$ is placed closer to $y_i$ than to $z_i$, i.e., $\|f(x_i) - f(y_i)\|_2 \leq \|f(x_i) - f(z_i)\|_2$. The embedding's objective value is $\sum_{i=1}^{m} w_i \cdot \mathbf{1}_{(x_i, y_i, z_i)}$, where $\mathbf{1}_{(x_i, y_i, z_i)} = 1$ if the constraint is satisfied by $f(\cdot)$, and 0 otherwise.

---
[a] The output embedding $f$ is computable in polynomial time in $|V|$ and the description of weights.

### LOCALCONTRASTIVE-TREE Problem.

INPUT : As above, set $V$ with contrastive triplets $\{(x_i, y_i, z_i)\}_{i=1}^{m}$ (non-negative weights $w_i \geq 0$).

OUTPUT : A hierarchical clustering, i.e., a binary rooted tree $T$ with $|V|$ leaves, and a 1-to-1 mapping from $V$ to the leaves of $T$, such that no vertex $v$ can increase the value of the tree by switching to another location in the tree $T$ (for each $v$ in $T$, there are exactly $2|V| - 3$ other candidate locations). We say a triplet $(x_i, y_i, z_i)$ is *satisfied* by $T$, if $x_i$ is placed closer to $y_i$ than to $z_i$, i.e., if there is a subtree in $T$ containing $x_i, y_i$ but not $z_i$ (this is equivalent to $\text{dist}_T(x_i, y_i) \leq \text{dist}_T(x_i, z_i)$ for an ultrametric distance on $T$). The tree's objective value is $\sum_{i=1}^{m} w_i \cdot \mathbf{1}_{(x_i, y_i, z_i)}$.

Moreover, we examine scenarios where provided contrastive information is of the form "among $x, y, z$, items $x, z$ are *farthest* apart" (instead of indicating that "item $x$ is closer to $y$ than to $z$"). Already for 1-dimensional embeddings, such constraints give rise to BETWEENNESS, a well-studied ranking problem in approximation algorithms [Arora et al., 2002, Charikar et al., 2009, Austrin et al., 2015].

### LOCALBETWEENNESS-EUCLIDEAN Problem.

INPUT : Set $V$ with *betweenness* triplets $\{(x_i, y_i, z_i)\}_{i=1}^{m}$ each with a non-negative weight $w_i \geq 0$, and target dimension $d$.

OUTPUT : An embedding $f : V \to \mathbb{R}^d$ such that[b] no vertex $v$ can increase the value of the embedding by switching its location in $\mathbb{R}^d$. We say a triplet $(x_i, y_i, z_i)$ is *satisfied* by $f(\cdot)$, if $x_i$ and $z_i$ are placed the farthest apart (equivalently, $y_i$ is "between" $x_i$ and $z_i$), i.e., $\|f(x_i) - f(z_i)\|_2 \geq \max\{\|f(x_i) - f(y_i)\|_2, \|f(z_i) - f(y_i)\|_2\}$. The embedding's objective value is $\sum_{i=1}^{m} w_i \cdot \mathbf{1}_{(x_i, y_i, z_i)}$.

---
[b] The output embedding $f$ is computable in polynomial time in $|V|$ and the description of weights.

Even though we defined problems in their general form, our PLS-hardness results also hold for interesting special cases: we show LOCALCONTRASTIVE-EUCLIDEAN in dimension $d = 1$ [Bădoiu et al., 2008, Alon et al., 2008, Fan et al., 2020] and LOCALBETWEENNESS-EUCLIDEAN with $d = 1$ [Arora et al., 2002, Charikar et al., 2009, Austrin et al., 2015] are PLS-hard. For trees, LOCALCONTRASTIVE-TREE is also known as triplet reconstruction/consistency [Byrka et al., 2010, Chatziafratis and Makarychev, 2023], and in terms of approximating the globally optimum solution several results and connections to other hierarchical clustering objectives are known [Chatziafratis et al., 2021, Naumov et al., 2021, Charikar et al., 2019, Chami et al., 2020, Moseley and Wang, 2023, Vainstein et al., 2021]. For other extensions, see Appendix C.

## 2.2 Continuous objectives in contrastive learning and CLS

Daskalakis and Papadimitriou [2011] proposed CLS to study local search for continuous objective functions, most notably search problems that can be solved by performing Gradient Descent. CLS has played an important role in game theory and optimization, and a recent breakthrough showed that CLS = PPAD ∩ PLS [Fearnley et al., 2023]. The natural problem of finding KKT points in quadratic programs was recently shown to be CLS-hard [Fearnley et al., 2024] and we will reduce it to finding local minima of the Triplet Loss [Schroff et al., 2015].

**QUADRATICPROGRAM-KKT Problem.**

INPUT : A symmetric matrix $\mathbf{Q} \in \mathbb{R}^{n \times n}$ and vector $b \in \mathbb{R}^n$.

OUTPUT : Compute a local optimum (i.e., a KKT point) for the quadratic $\min_{\boldsymbol{x} \in [0,1]^n} \boldsymbol{x}^\top \mathbf{Q} \boldsymbol{x} + \boldsymbol{b}^\top \boldsymbol{x}$.

**LOCALTRIPLETLOSS-EUCLIDEAN Problem.**

INPUT : Set $V \cup \{A, B\}$ of points in $\mathbb{R}^d$, set $\mathcal{C}$ of triplets $(\boldsymbol{x}, \boldsymbol{y}, \boldsymbol{z})$ of weight $w \geq 0$, margin $\alpha > 0$.

OUTPUT : Find[c] an embedding $f : V \to [0, 1]^d$ that is a first-order stationary point for the minimization objective given by the triplet-loss:

$$\sum_{(\boldsymbol{x}, \boldsymbol{y}, \boldsymbol{z}) \in \mathcal{C}} w \cdot \max \left\{ \|f(\boldsymbol{x}) - f(\boldsymbol{y})\|_2^2 - \|f(\boldsymbol{x}) - f(\boldsymbol{z})\|_2^2 + \alpha, 0 \right\}.$$

[c]The output embedding $f$ is computable in polynomial time in the description of the input set $V$ and the weights.

Recall, first-order stationary points are fixed points of gradient descent: $\boldsymbol{x}^* \in \mathcal{X}$ is a first-order stationary point of $g(\boldsymbol{x})$, if $\forall \boldsymbol{x} \in \mathcal{X}$ we have $\langle \boldsymbol{x} - \boldsymbol{x}^*, \nabla_{\boldsymbol{x}} g(\boldsymbol{x}^*) \rangle \geq 0$ ($\mathcal{X}$: convex, compact domain).

We emphasize the role of the two pivot points $A, B$ in the definition. Notice that if we did not have pivot points, then the embedding that maps all points from $V$ to the all-zeros vector would be a trivial first-order stationary point. Moreover, if we had only 1 pivot $A$, then again mapping every point in $V$ to $A$ would result in a first-order stationary point of the Triplet Loss. Thus having two pivot points are necessary and sufficient to make the problem non-trivial.

## 3 PLS-hardness for discrete objectives in contrastive learning

In this section we present our results for LOCALCONTRASTIVE-EUCLIDEAN, LOCALCONTRASTIVE-TREE, LOCALBETWEENNESS-EUCLIDEAN where the goal is to find local solutions for maximizing weight of satisfied triplets. We start with Euclidean embeddings, then have results on trees.

### 3.1 The case of embeddings in $\mathbb{R}^d$ with $d = 1$

To set some notation and convey the key ideas for our later proofs, we start by presenting our reductions for the case of 1-dimensional embeddings, since our reductions for the general case are more involved.

**Theorem 3.1.** *LOCALCONTRASTIVE-EUCLIDEAN, even for embedding dimension $d = 1$, is* PLS*-hard.*

*Proof.* We present a PLS-reduction from LOCALMAXCUT to LOCALCONTRASTIVE-EUCLIDEAN with $d = 1$. Let $G = (V, E)$ be an undirected graph with edge-weights $w \geq 0$. Let $W := \sum_{(u,v) \in E} w_{uv}$ be the total weight. Moreover, let $M := W + 1$ and $M' := (2|V| + 1)M$ denote two heavy weights. For our reduction, we introduce three special vertices $X, Y, Z$ (we use capital letters to distinguish them from the vertices of $G$). Eventually, the input to the LOCALCONTRASTIVE-EUCLIDEAN problem will be $V \cup \{X, Y, Z\}$ together with a collection of $m = |E| + 2|V| + 2$ contrastive triplets. Our goal is to show how we can recover a locally maximum cut for $G$, from a locally maximum embedding of our constructed instance.

We add the following two types of contrastive triplet constraints:

- **Type A ("edge" triplets).** For every edge $(u, v) \in E$ with weight $w_{uv}$, add a single triplet $(u, X^+, v^-)$ with weight $w_{uv}$. This incentivizes embeddings to put $u$ closer to $X$ than $v$.

- **Type B ("boundary" triplets).** We add the following:
  - For every vertex $v \in V$, add $(X, Y^+, v^-)$ with weight $M$.
  - For every vertex $v \in V$, add $(X, v^+, Z^-)$ with weight $M$.
  - Finally, add $(Y, Z^+, X^-)$ with weight $M'$, and $(X, Y^+, Z^-)$ also with weight $M'$.

In total, this creates $m = |E| + 2|V| + 2$ contrastive triplets on $V \cup \{X, Y, Z\}$.

First, observe that because of the heavy-weight triplets $(X, Y^+, Z^-)$ and $(Y, Z^+, X^-)$, any locally maximum embedding must satisfy those two triplets: if any of those two triplets was violated, we can always satisfy both simultaneously by moving $Z$ and forcing either the order $X < Y < Z$ or

$Z < Y < X$, with distances $|X - Y| \geq |Y - Z|$, thus gaining at least $M'$ while losing at most $2|V|M + \sum_{(u,v) \in E} w_{uv} < M'$, which would contradict local optimality.

Next, we show that in any locally maximum embedding, all $(X, Y^+, v^-)$ and $(X, v^+, Z^-)$ are satisfied, meaning that each $v \in V$ must lie on one of two line segments $YZ$ or $Y'Z'$, where $Y'$ and $Z'$ are the reflections of $Y$ and $Z$ with respect to $X$ respectively (see Figure 2). Note that $Y', Z'$ are only used for the analysis and they are not part of the reduction. If any $(X, Y^+, v^-)$ or $(X, v^+, Z^-)$ is violated, we can always satisfy both by moving $v$ to segment $YZ$ or to $Y'Z'$, thus gaining at least $M$ while losing at most $\sum_{(u,v) \in E} w_{uv} < M = \sum_{(u,v) \in E} w_{uv} + 1$.

$$Z' \quad Y' \qquad\qquad X \qquad\qquad Y \quad Z$$

Figure 2: Reduction ($d = 1$): Any local optimum places $v \in V$ in segment $YZ$ or its reflection $Y'Z'$.

Finally, it is clear that $(u, X^+, v^-)$ is satisfied if and only if $u$ and $v$ are placed on different sides of $X$. This encodes a solution to the LOCALMAXCUT instance, by defining one side of the cut $(S, \overline{S})$ to be all vertices placed in segment $YZ$ and the other side to be the remaining vertices. $\qquad\square$

## 3.2 Hardness for general $d$-dimensional embeddings

Extending the above proof, our two results for general Euclidean embeddings are:

**Theorem 3.2.** *For every fixed dimension $d \geq 1$, LOCALCONTRASTIVE-EUCLIDEAN is PLS-hard.*

**Theorem 3.3.** *For every fixed dimension $d \geq 1$, LOCALBETWEENNESS-EUCLIDEAN is PLS-hard.*

All proofs can be found in the Appendix A. Due to space constraints, we give the proof for LOCALBETWEENNESS-EUCLIDEAN for the case $d = 2$. Similar gadgets are used to prove Theorem 3.2.

*Proof of Theorem 3.3 ($d = 2$).* Our PLS-reduction is from LOCALMAXCUT. As previously, let $G = (V, E)$ be an undirected graph with edge-weights $w \geq 0$. Let $W := \sum_{(u,v) \in E} w_{uv}$, and let $M := W + 1$ denote a heavy weight. For our reduction, we introduce two special vertices $X_1, X_2$. Our goal is to show how we can recover a locally maximum cut for $G$, from a locally maximum embedding of our constructed instance. We add the following two types of contrastive triplet constraints:

- **Type A ("edge" triplets).** For each edge $(u, v) \in E$ with weight $w_{uv}$, add a single triplet $(u, X_1, v)$ with weight $w_{uv}$. Geometrically, the semantics of the triplet in LOCALBETWEENNESS-EUCLIDEAN is that the pair $u, v$ is the farthest distance apart, i.e., the largest edge in the triangle formed by $u, X_1, v$ is the edge $(u, v)$.

- **Type B ("isosceles" triplets).** For each $v \in V$, add two triplets $(X_1, X_2, v)$ and $(X_2, X_1, v)$, each with weight $M$. Intuitively, because of the heavy weight, this forces $X_1, X_2, v$ to form an *isosceles triangle*, such that $\|v - X_1\|_2 = \|v - X_2\|_2 \geq \|X_1 - X_2\|_2$, where we overload the notation $v, X_1, X_2$ to also denote the vertex embeddings in the 2D-plane.

Observe that in any local optimum, all "isosceles" triplets must be satisfied. If any "isosceles" triplet involving $v$ is unsatisfied, we can always satisfy it by moving $v$ to form the corresponding isosceles triangle, thus gaining at least $M$ while losing at most $\sum w_{uv} < M$, which would strictly increase the objective value. This effectively forces every vertex $v$ to lie on one of two rays (see Figure 3).

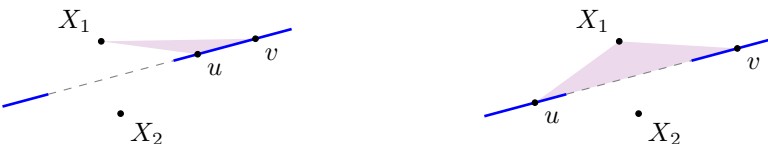

Figure 3: Reduction ($d = 2$) for LOCALBETWEENNESS-EUCLIDEAN: Type B triplets force all $v \in V$ onto two opposing rays. Left: $(u, X_1, v)$ is not satisfied when $u$ and $v$ lie on the same ray. Right: $(u, X_1, v)$ is satisfied when $u$ and $v$ lie on different rays.

Next, we show that in any locally maximum embedding for LocalBetweenness-Euclidean, an "edge" triplet $(u, X_1, v)$ is satisfied if and only if $u$ and $v$ are on *opposite* rays. To see this, let us analyze the triangle $\triangle u X_1 v$:

- If $u$ and $v$ are on the same ray, we have $\angle u X_1 v < 30°$. Basic trigonometry tells us $uv$ cannot be the longest side of the triangle, thus $(u, X_1, v)$ is not satisfied.
- If $u$ and $v$ are on opposite rays, we have $\angle u X_1 v \geq 120°$. Basic trigonometry tells us $uv$ must be the longest side of the triangle, thus $(u, X_1, v)$ is satisfied.

Therefore, the configuration of a locally maximum embedding encodes a local max cut for $G$. We can easily recover the cut $(S, \overline{S})$ by checking whether each "edge" triplet $(u, X_1, v)$ is satisfied in the configuration, e.g., $S$ can be all vertices in one ray. $\qquad\square$

### 3.3 Hardness of tree embeddings

**Theorem 3.4.** *The LocalContrastive-Tree problem is* PLS-*hard.*

*Proof Sketch.* We do a PLS-reduction from LocalMaxCut. Given graph $G(V, E)$, we construct triplets over nodes $V \cup \{X, X', Y, Z\}$ for special nodes $\{X, X', Y, Z\}$. The triplets are as follows:

- **Type A triplets**: $XX'|Y$, $XX'|Z$, and $XX'|v$ for all $v \in V$, each with weight $\sum_{e \in E} w_e$.
- **Type B triplets**: $XY|Z$ with a large weight $|V| \cdot \sum_{e \in E} w_e + |V| + 1$.
- **Type C triplets**: $Yv|X$ and $Zv|X$ for all $v \in V$, each with weight $\sum_{e \in E} w_e + 1$.
- **Type D triplets**: $uX|v$ and $vX|u$ whenever $(u, v) \in E$, each with weight $w_{uv}/2$.

The purpose of the special vertices and the constraints is that they force the tree to look as in Figure 4: $X, X'$ are siblings, and any other vertex $v \in V$ is either in subtree $T_Y$ containing $Y$ or subtree $T_Z$ containing $Z$ (because of heavy Type C triplets). Then, the solution to LocalMaxCut can be formed by all nodes that fell in $T_Y$ as one side of the cut, and to $T_Z$ as the other side. The proof proceeds by a case analysis showing that this is indeed a local max cut, and that no vertex can increase the weight of the cut edges by moving to the other subtree (for full proof, see Appendix B). $\qquad\square$

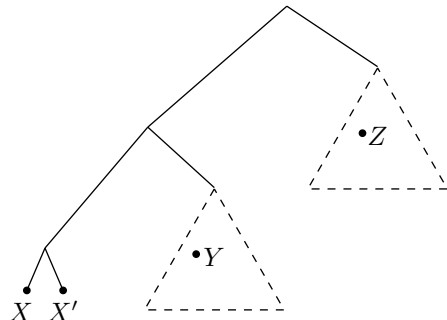

Figure 4: Reduction for LocalContrastive-Tree.

## 4    CLS-**hardness for continuous objectives in contrastive learning**

**Theorem 4.1.** *For every dimension $d \geq 1$, LocalTripletLoss-Euclidean is* CLS-*hard.*

*Proof Sketch.* We present the key idea for the case $d = 1$ with two pivot points $A = 0$ and $B = 1/2$. The full proof is in Appendix D. We give a CLS-reduction from QuadraticProgram-KKT. The key idea is that we can use triplets of the form $(x, y, z)$ to simulate the quadratic function. For now, imagine we had as a representation function $f(\cdot)$ the identity function. Then, a given triplet constraint $(x, y, z)$ with weight $w$ will contribute the following term to the overall Triplet Loss:

$$\mathcal{L}(x, y, z) = w \cdot \max\{(x - y)^2 - (x - z)^2 + \alpha, 0\}.$$

Our proof first breaks the given quadratic program into smaller quadratics on three variables at a time:

$$q(x, y, z) := c_1 x^2 + c_2 y^2 + c_3 z^2 + c_4 xy + c_5 xz + c_6 yz + c_7 x + c_8 y + c_9 z.$$

We then show how to generate a set of $m$ carefully chosen triplets $\{(x_i, y_i, z_i)\}_{i=1}^m$ with appropriate weights $w_i$ so that the total triplet loss $\mathcal{L}$ is the same as the quadratic program. It is easy to see that for terms depending only on one variable such as $c_1 x^2$ or $c_8 y$, we can easily generate them by using a triplet (for example, triplet $(0, x, \frac{1}{2})$ allows us to generate the quadratic term $x^2$ and triplet $(y, 0, \frac{1}{2})$ generates the linear term $y$). However, the main obstacle is that for the cross-terms like $xy$ we need to use triplet $(x, 0, y)$, and this introduces dependencies among the different triplets since there are shared variables. To overcome the difficulty of interacting terms, we need to introduce a total of 12 contrastive triplets on $x, y, z$, whose corresponding weights depend on the coefficients of the given quadratic. By solving a linear system for the weights, we obtain a loss $\mathcal{L}(x, y, z)$ that equals $q(x, y, z)$.

Using these ideas, we can prove hardness under the framework of contrastive learning [Schroff et al., 2015], even for the case where the representation $f(\cdot)$ is a linear model parameterized by $\boldsymbol{\theta} \in [0, 1]^n$ such that $f_{\boldsymbol{\theta}}(\boldsymbol{x}) = \boldsymbol{\theta}^\top \boldsymbol{x}$. Here $\boldsymbol{x}$ is the sampled input to the linear model, $f(\boldsymbol{x})$ is the output of the linear model. For any input $\boldsymbol{x}_i$, denote $a_i(\boldsymbol{\theta}) = \boldsymbol{\theta}^\top \boldsymbol{x}_i$ to be the output of the linear model. We aim to find local solutions of the triplet loss problem:

$$\min_{\boldsymbol{\theta} \in [0,1]^n} \sum_{(i,j,k)} w_{i,j,k} \max\left( (a_i(\boldsymbol{\theta}) - a_j(\boldsymbol{\theta}))^2 - (a_i(\boldsymbol{\theta}) - a_k(\boldsymbol{\theta}))^2 + \alpha, 0 \right).$$

We set $\alpha = 1$ and let $\boldsymbol{x}_i = \boldsymbol{e}_i$ to be the $i^{th}$ standard basis, then $a_i(\boldsymbol{\theta}) = \theta_i \in [0, 1]$. The result follows from the case with $d = 1$. □

## 5 Experimental verification: hard examples for local search

Our theoretical results establish worst-case hardness for finding local optima in several contrastive objectives. Here, we provide an experimental illustration that this indeed can happen by constructing instances where local search takes exponential time. By using our reductions from LocalMaxCut, we create hard instances for various other contrastive problems, namely LocalContrastive-Euclidean, LocalContrastive-Tree and LocalBetweenness-Euclidean, and we measure the runtime of local search.

Our starting point is a hard instance for the LocalMaxCut problem due to Monien and Tscheuschner [2010]. This instance is a bounded-degree graph with maximum vertex degree 4, where any flip-based local search algorithm (initialized from a specific initial cut) will require exponentially many iterations to reach a local optimum. Flip-based local search algorithms attempt to move one vertex at a time from its current position to another position, while trying to improve the objective. Our reductions transform this instance to the various contrastive objectives mentioned previously. In fact, our reductions when applied to the hard instance of LocalMaxCut preserve the local search structure and the changes in the objectives exactly. As we verify, the same exponentially-slow path towards a local optimum exists in all of the transformed instances. In particular, to validate our findings, we implemented local search for LocalMaxCut, as well as for the three problems LocalContrastive-Euclidean, LocalBetweenness-Euclidean, and LocalContrastive-Tree. Interestingly, in all four cases, we observed exactly the same local search dynamics (i.e., the same improving moves were performed). As a consequence the iteration counts on the corresponding instances are identical. We summarize the results in Figure 5 and provide the detailed statistics in Appendix E.

Finally, we observed that if we start from a random configuration, local search usually converges quickly, suggesting that the hard instances of Monien and Tscheuschner [2010] are sensitive to the choice of the starting point. However, understanding whether random initialization (or other strategies) is generally sufficient to avoid such worst-case dynamics remains an open question.

## Conclusion

In this work, we provided strong evidence that computing local optima for various contrastive learning objectives is computationally intractable in the worst case. To do so we relied on the well-studied complexity classes PLS and CLS and we presented a series of reductions from difficult problems

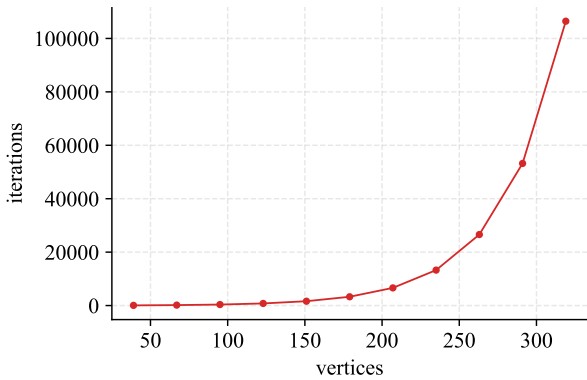

Figure 5: Comparison of instance sizes and number of iterations of local search for the sequence of hard instances created by our reductions.

of those classes to our contrastive objectives (even for relatively simple settings). There are many exciting questions for future consideration. The main interesting future direction is to examine whether our negative results are persistent in the average case. In particular, understanding the effects of initialization is an exciting direction. Another interesting perspective to study for contrastive objectives, is so-called smoothed analysis [Spielman and Teng, 2009], where the input data are slightly perturbed by random noise, and many algorithms (including linear programming with the Simplex, which is a local search algorithm) seem to get much better guarantees compared to worst-case inputs. Finally, our work studied the question of how fast can we reach a local optimum, but an important future direction is to argue about the *quality* of local solutions.

## Acknowledgements

Ioannis Panageas and Jingming Yan were supported by NSF grant CCF-2454115. Vaggos Chatziafratis is supported by a Hellman Fellowship and a startup grant from UC Santa Cruz. This work has been partially supported by project MIS 5154714 of the National Recovery and Resilience Plan Greece 2.0 funded by the European Union under the NextGenerationEU Program. Part of this work was done while J.Y., Y.L., V.C., and I.P. were visiting Archimedes AI Research Center, Athens, Greece.

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

# A  Omitted proofs for Section 3.2

**Theorem A.1** (Theorem 3.3, restated). *For every fixed dimension $d \geq 1$, LOCALBETWEENNESS-EUCLIDEAN is* PLS-*hard.*

*Proof.* We reduce from LOCALMAXCUT. Let $G = (V, E, w)$ be a weighted undirected graph.

**Reduction Construction.** We define an instance of LOCALBETWEENNESS-EUCLIDEAN in $\mathbb{R}^d$ called $\mathcal{I}_{btw}$ as follows. We introduce special vertices $X_1, \ldots, X_d$ and choose a hierarchy of weights

$$M_3 > M_4 > \cdots > M_d > M > \sum_{(u,v) \in E} w_{uv},$$

such that for each $k \geq 3$,

$$M_k > \underbrace{\sum_{j > k} 3\binom{j-1}{2} M_j}_{\text{lower layer "equilateral" triplets}} + \underbrace{2|V|\binom{d}{2} M}_{\text{all "isosceles" triplets}} + \underbrace{\sum_{(u,v) \in E} w_{uv}}_{\text{all "edge" triplets}}.$$

Now we add constraints as follows:

- **Type A ("edge" triplets).** For each edge $(u, v) \in E$, we add a triplet $(u, X_1, v)$ with weight $w_{uv}$.

- **Type B ("isosceles" triplets).** For each $v \in V$ and every pair $(X_k, X_\ell)$ with $1 \leq k < \ell \leq d$, we add two triplets $(X_k, X_\ell, v)$ and $(X_\ell, X_k, v)$, each with weight $M$. Intuitively, this will force $X_k, X_\ell, v$ to form an *isosceles triangle* such that $\|v - X_k\|_2 = \|v - X_\ell\|_2 \geq \|X_k - X_\ell\|_2$.

- **Type C ("equilateral" triplets).** For every triple $(X_j, X_k, X_\ell)$ with $1 \leq j < k < \ell \leq d$, we add three triplets $(X_j, X_k, X_\ell), (X_k, X_\ell, X_j), (X_\ell, X_j, X_k)$, each with weight $M_\ell$ ($\ell$ is the largest index among $j, k, \ell$). Intuitively, this will force $X_j, X_k, X_\ell$ to form an *equilateral triangle*.

**Lemma A.2.** *In any local optimum of $\mathcal{I}_{btw}$, all "equilateral" triplets are satisfied, therefore $X_1, \ldots, X_d$ form a $(d-1)$-dimensional regular simplex.[‡]*

*Proof of Lemma A.2.* We prove $\{X_1, \ldots, X_k\}$ form a regular simplex for each $k = 3, \ldots, d$. We do this by induction on $k$.

**Base Case ($k = 3$).** If any "equilateral" triplets over $\{X_1, X_2, X_3\}$ is unsatisfied, moving any of them to form an equilateral triangle with the other two will satisfy at least one new triplet of weight $M_3$, while losing at most $\sum_{j > 3} 3\binom{j-1}{2} M_j + 2|V|\binom{d}{2} M + \sum_{(u,v) \in E} w_{uv} < M_3$, which contradicts local optimality.

**Inductive Step.** Suppose $\{X_1, \ldots, X_{k-1}\}$ form a regular simplex but $\{X_1, \ldots, X_k\}$ do not. Similar to the above, moving $X_k$ to form a regular simplex with $\{X_1, \ldots, X_{k-1}\}$ will satisfy at least one new triplet of weight $M_k$, while losing a total strictly less than $M_k$. Hence, $\{X_1, \ldots, X_k\}$ must also form a regular simplex. $\square$

**Lemma A.3.** *Let $C$ be the centroid of the simplex $X_1, \ldots, X_d$ and $\ell = \|X_1 - X_2\|_2$ be its edge length. Then in any local optimum, all "isosceles" triplets are satisfied, therefore each $v \in V$ must lie on the line through $C$ perpendicular to the hyperplane spanned by the $X_i$, and moreover at distance at least $\ell$ from every $X_i$. Formally, each $v \in V$ lies on one of two opposing rays:*

$$R^+ = \big\{ C + t\, \mathbf{u} : t \geq t_0 \big\}, \quad R^- = \big\{ C - t\, \mathbf{u} : t \geq t_0 \big\},$$

*where $\mathbf{u}$ is the unit vector along that line and $t_0 = \ell \cdot \sqrt{\frac{d+1}{2d}}$.*

*Proof of Lemma A.3.* Without loss of generality, assume the simplex is in the standard position:

$$X_1 = (1, 0, \ldots, 0), X_2 = (0, 1, \ldots, 0), \ldots, X_d = (0, 0, \ldots, 1)$$

---

[‡] In geometry, a regular simplex is the $d$-dimensional generalization of an equilateral triangle (2-simplex) or a regular tetrahedron (3-simplex), consisting of $d + 1$ points with all pairwise distances equal.

with centroid $C = \left(\frac{1}{d}, \frac{1}{d}, \ldots, \frac{1}{d}\right)$. The edge length is $\ell = \sqrt{2}$.

We denote the $i$-th coordinate of $v$ as $v_i$. The equidistant condition $\|v - X_k\|_2 = \|v - X_\ell\|_2$ simplifies to:
$$v_i^2 - 2v_i = v_j^2 - 2v_j \quad \forall i \neq j.$$

The only solutions are $v = (\alpha, \alpha, \ldots, \alpha)$ for some $\alpha \in \mathbb{R}$.

For $v = \alpha\mathbf{1}$, the distance to any vertex $X_k$ is:
$$\|v - X_k\|_2^2 = (\alpha - 1)^2 + (d - 1)\alpha^2 = d\alpha^2 - 2\alpha + 1 \geq \ell^2 = 2.$$

Solving $d\alpha^2 - 2\alpha + 1 \geq 2$ gives:
$$\alpha \leq \frac{1 - \sqrt{d + 1}}{d} \quad \text{or} \quad \alpha \geq \frac{1 + \sqrt{d + 1}}{d}.$$

Expressing $v$ relative to the centroid $C = \frac{1}{d}\mathbf{1}$:
$$v = C + t\,\mathbf{u}, \quad \mathbf{u} = \mathbf{1}/\sqrt{d}, \quad t = \sqrt{d}\left(\alpha - \frac{1}{d}\right).$$

Substituting the bounds for $\alpha$ yields $t \leq -\sqrt{\frac{d+1}{d}}$ or $t \geq \sqrt{\frac{d+1}{d}}$.

For arbitrary $\ell > 0$, scale the canonical simplex by $\lambda = \ell/\sqrt{2}$. The threshold $t_0$ scales as:
$$t_0 = \lambda\sqrt{\frac{d+1}{d}} = \ell\sqrt{\frac{d+1}{2d}}.$$

The standard basis assumption is justified because any regular simplex can be mapped to this form via a similarity transformation, which preserves the distance relationships.

Since $M > \sum_{(u,v)\in E} w_{uv}$, any configuration of $v$ that violates the "isosceles" triplets can be improved by moving $v$ to either $R^+$ or $R^-$. $\qquad\square$

**Lemma A.4.** *In any local optimum, the "edge" triplet $(u, X_1, v)$ is satisfied if and only if $u$ and $v$ lie on opposite rays $R^+$ and $R^-$.*

*Proof of Lemma A.4.* Let $u = C + t_i\mathbf{u}$ and $v = C + t_j\mathbf{u}$. Since $u, v$ lie on the line through $C$ perpendicular to the hyperplane spanned by the $X_i$, we have
$$\|u - X_1\|_2 = \sqrt{t_i^2 + \|X_1 - C\|_2^2},$$
$$\|v - X_1\|_2 = \sqrt{t_j^2 + \|X_1 - C\|_2^2}.$$

We can also verify that $\|X_1 - C\|_2 = \ell\sqrt{\frac{d-1}{2d}}$.

**Case 1:** If $u$ and $v$ are on the same ray (e.g., $R^+$), then:
$$\|u - v\|_2 = |t_i - t_j| \leq \max(t_i, t_j) < \max(\|u - X_1\|_2, \|v - X_1\|_2).$$

Thus $(u, X_1, v)$ is not satisfied.

**Case 2:** If $u$ and $v$ are on opposite rays $R^+$ and $R^-$, then:
$$\|u - X_1\|_2 = \sqrt{t_i^2 + \|X_1 - C\|_2^2} = \sqrt{t_i^2 + \frac{(d-1)\ell^2}{2d}},$$
$$\|v - X_1\|_2 = \sqrt{t_j^2 + \|X_1 - C\|_2^2} = \sqrt{t_j^2 + \frac{(d-1)\ell^2}{2d}},$$
$$\|u - v\|_2 = |t_i| + |t_j|.$$

We need to show that $\|u - v\|_2 \geq \max(\|u - X_1\|_2, \|v - X_1\|_2)$.

To show $\|u - v\|_2 \geq \|u - X_1\|_2$, we need $(|t_i| + |t_j|)^2 \geq t_i^2 + \frac{(d-1)\ell^2}{2d}$. Expanding:

$$t_i^2 + 2|t_i t_j| + t_j^2 \geq t_i^2 + \frac{(d-1)\ell^2}{2d} \implies 2|t_i t_j| + t_j^2 \geq \frac{(d-1)\ell^2}{2d}.$$

Since $|t_i|, |t_j| \geq t_0$, substitute $|t_i| = |t_j| = t_0$ (worst case for the inequality):

$$2t_0^2 + t_0^2 = 3t_0^2 \geq \frac{(d-1)\ell^2}{2d}.$$

Substitute $t_0 = \ell\sqrt{\frac{d+1}{2d}}$:

$$3 \cdot \frac{(d+1)\ell^2}{2d} \geq \frac{(d-1)\ell^2}{2d} \implies 3(d+1) \geq d - 1.$$

This holds for all $d \geq 2$.

Symmetrically we can show $\|u - v\|_2 \geq \|v - X_1\|_2$. Thus $(u, X_1, v)$ is satisfied. $\square$

**Special Case** $d = 1$. We note that in the case of $d = 1$, this construction degenerates into only one special vertex $X_1$, with only "edge" triplets. We claim that this construction is still correct, as the vertices on different sides of $X_1$ encode a cut. Since "no local moves can improve the objective function" is a stronger condition than "no local moves of vertices except $X_1$ can improve the objective function", any local optimum of $\mathcal{I}_{btw}$ is also a local max cut.

This concludes the proof of Theorem 3.3. $\square$

*Proof of Theorem 3.2.* We extend the reduction for LOCALCONTRASTIVE-EUCLIDEAN to higher dimensions by modifying the LOCALBETWEENNESS-EUCLIDEAN construction.

Recall that in the LOCALBETWEENNESS-EUCLIDEAN construction, all $v \in V$ are forced onto two opposing rays. We can encode the same isosceles and equilateral gadgets using contrastive triplets, for example, replacing a betweenness triplet $(x, y, z)$ with two contrastive triplets $(x, y^+, z^-)$ and $(z, y^+, x^-)$. We then use the same idea for the $d = 1$ case we considered in Section 3.1, to turn the rays into segments. Let $Y$ be a new special vertex forming a regular simplex with $X_1, \ldots, X_d$, and $Z$ be a new special vertex on one of the rays. We add a contrastive triplet $(Y, Z^+, X_1^-)$ to ensure that $\|X_1 - Y\|_2 \geq \|Y - Z\|_2$, so $YZ$ is the segment we "cut" on the ray.

For each $v \in V$, we add two contrastive triplets $(X_1, Y^+, v^-)$ and $(X_1, v^+, Z^-)$ to ensure that $v$ is on $YZ$, or on its mirror $Y'Z'$ with respect to the hyperplane spanned by $X_1, \ldots, X_d$.

We can always choose a hierarchy of weights so that these newly added triplets are satisfied in any local optimum. The rest of the proof follows similarly to Theorem 3.3. $\square$

# B  Omitted proofs for Section 3.3

*Proof of Theorem 3.4.* We reduce from LOCALMAXCUT. For a given graph $G(V, E, w)$, we construct the triplets over vertices $V \cup \{X, X', Y, Z\}$. We denote $W := \sum_{(u,v) \in E} w_{uv}$ to be the sum of edge weights. The triplets are constructed as

- **Type A triplets**: $XX'|Y$, $XX'|Z$, and $XX'|v$ for all vertex $v \in V$, each with weight $W$;

- **Type B triplets**: $XY|Z$ with a large weight $nW + n + 1$;

- **Type C triplets**: $Yv|X$ and $Zv|X$ for all $v \in V$, each with weight $W + 1$;

- **Type D triplets**: For every edge $(u, v) \in E$, we add $uX|v$ and $vX|u$, each with weight $w_{uv}/2$.

Assume that $T$ is a local solution of the above triplets instance. Since $T$ is an rooted binary tree, there exists a unique path $P$ starting from the leave node $X$ to the root. All other leave nodes $V \cup \{X', Y, Z\}$ would be on branches of this path $P$. Firstly observe that $X'$ should share the same parent node with $X$. In other words, $X'$, and only $X'$, should be on the lowest branch on the path $P$. If this is not the case, $X'$ can simply move to a location such that $X$ and $X'$ share the same parent and this would only increase the satisfied triplets by satisfying all triplets of Type A.

Since in any local solution $T$, Type A triplets are always satisfied as argued above. For nodes $Y$ and $Z$, Type B triplet $XY|Z$ has dominating weight (the weight of $XY|Z$ is greater than the sum of weights of other triplets that $Y$ or $Z$ is involved). Formally, we have

$$w_{XY|Z} > \sum_{v \in V} w_{Yv|X} \quad \text{and} \quad w_{XY|Z} > \sum_{v \in V} w_{Zv|X}.$$

Thus any local solution $T$ needs to satisfy Type B triplet. We note that the triplet $XY|Z$ can only be satisfied when $Y$ is on a lower branch of $Z$ on the path $P$.

Similarly for each vertex $v \in V$, it holds that

$$w_{Yv|X} > \sum_{u \in \mathcal{N}(v)} \left( w_{uX|v} + w_{vX|u} \right)$$

where $\mathcal{N}(v)$ denotes the set of neighbors of $v$. Thus in any local solution $T$, vertex $v$ would try to satisfy $Yv|X$ and $Zv|X$ first. Since $Y$ and $Z$ are on different branch, only one of $Yv|X$ and $Zv|X$ could be satisfied in a local solution $T$. Notice that in order to satisfy $Yv|X$ or $Zv|X$, vertex $v$ should be on the branch of $Y$ or the branch of $Z$ respectively. We conclude that in a local solution $T$, any $v \in V$ should be on either the branch with $Y$ or the branch with $Z$.

We proceed to show that for any pair of vertices $u$ and $v$, if $u$ and $v$ are on the same branch (either the branch with $Y$ or the branch with $Z$), then neither of $vX|u$ or $uX|v$ would be satisfied. If $u$ and $v$ are on different branches, e.g., $v$ is on the branch of $Y$ and $u$ is on the branch of $Z$, then exactly one of $vX|u$ or $uX|v$ would be satisfied. Since any vertex $v \in V$ can only reside in either the branch with $Y$ or the branch with $Z$, we conclude that in any local solution $T$, the configuration of all vertices $v$ will introduce a solution of LOCALMAXCUT on the original graph. That is, all vertices $v \in V$ on the branch with $Y$ form one side of the cut and those on the branch with $Z$ form the other side. $\qquad\square$

## C   Hardness of Non-Betweeness embeddings

We start this section by giving formal definition of LOCALNONBETWEENNESS-EUCLIDEAN problem.

> **LOCALNONBETWEENNESS-EUCLIDEAN Problem.**
> INPUT : Set $V$ with *non-betweenness* triplets $\{(x_i, y_i, z_i)\}_{i=1}^m$ each with a non-negative weight $w_i \geq 0$, and target dimension $d$.
> OUTPUT : An embedding[a] $f : V \to \mathbb{R}^d$ such that no vertex $v$ can increase the value of the embedding by switching its location in $\mathbb{R}^d$. We say a triplet $(x_i, y_i, z_i)$ is *satisfied* by $f(\cdot)$, if $x_i$ and $z_i$ are not placed the farthest apart (equivalently, $y_i$ is not "between" $x_i$ and $z_i$), i.e., $\|f(x_i) - f(z_i)\|_2 \leq \max\{\|f(x_i) - f(y_i)\|_2, \|f(z_i) - f(y_i)\|_2\}$. The embedding's objective value is $\sum_{i=1}^m w_i \cdot \mathbf{1}_{(x_i, y_i, z_i)}$.
>
> ---
> [a]The output embedding $f$ is computable in polynomial time in the description of the input set $V$ and the weights.

The problem LOCALNONBETWEENNESS-EUCLIDEAN problem, even in the case of 1-dimensional embeddings ($d = 1$), it is the well-studied problem called Non-Betweenness [Guruswami et al., 2008, Charikar et al., 2009, Austrin et al., 2015]. Here we show that it is PLS-hard.

**Theorem C.1.** *LOCALNONBETWEENNESS-EUCLIDEAN for embedding dimension $d = 1$ is PLS-hard.*

*Proof.* We give a polynomial time reduction from LOCALMAXCUT to LOCALNONBETWEENNESS-EUCLIDEAN with $d = 1$. Given an undirected graph $G = (V, E)$ with edge-weights $w \geq 0$. Denote $W := \sum_{(u,v) \in E} w_{uv}$ to be the sum of all edge weights. We introduce two special vertices $X$ and $Y$ and the input to the LOCALNONBETWEENNESS-EUCLIDEAN will be vertices from $V \cup \{X, Y\}$ with $|V| + 2|E|$ non-betweenness triplets. The triplets can be classified into two types:

- **Type A triplets**. For every vertex $v \in V$, we add a triplet $(X, v, Y)$ with weight $W$.

- **Type B triplets**. For every edge $(u, v) \in E$, we add $(u, v, X)$ and $(v, u, X)$ with weight $w_{uv}$.

We now argue that in any local optimal embedding over $\mathbb{R}$, there must be no point between $X$ and $Y$. To see this, given an embedding where vertex $v$ is in between $X$ and $Y$, the corresponding triplet

$(X, v, Y)$ is not satisfied. By moving $Y$ to the location such that there is no point between $X$ and $Y$ ($X$ and $Y$ are next to each other), triplet $(X, v, Y)$ becomes satisfied and all previous satisfied triplets remain unchanged. Thus one can simply change the location of $Y$ and increase the sum of weighted satisfied triplets.

We proceed to analyze Type B triplets. For any edge $(u, v) \in E$, we have the following three cases.

**Case 1** Vertices $u$ and $v$ are to the left of $X$, that is, $u$ and $v$ are both smaller than $X$. Without loss of generality, we assume that $u < v$. Notice that in this case $v$ is in between $u$ and $X$. Thus triplet $(v, u, X)$ is satisfied but $(u, v, X)$ is not satisfied.

**Case 2** Both $u$ and $v$ are greater than $X$. Assume that $u < v$, it follows that triplet $(u, v, X)$ is satisfied but $(v, u, X)$ is not satisfied.

**Case 3** Vertices $u$ and $v$ are on different side of $X$. In this case it holds that

$$|u - X| \leq |u - v|, \quad \text{and} \quad |v - X| \leq |u - v|.$$

Thus both triplets $(u, v, X)$ and $(v, u, X)$ are satisfied.

We conclude that for any edge $(u, v) \in E$, if vertices $u$ and $v$ are on the same side of $X$ (either to the left or to the right), then one of triplets $(u, v, X)$ or $(v, u, X)$ would be satisfied. If $u$ and $v$ are on different side of $X$, both $(u, v, X)$ and $(v, u, X)$ would be satisfied. Thus type B triplets essentially induce a cut over the original graph $G$–any vertex $v$ to the left of $X$ is on one side of the cut and vertices to the right of $X$ is on the other side of the cut. Any local optimal embedding over the above LocalNonBetweenness-Euclidean instance with total weight $W'$ would induce a LocalMaxCut over $G$ with weight $W' - (|V| + 1)W$. □

## D    Omitted proofs for Section 4

*Proof of Theorem 4.1.* We first study the case where the embedding dimension $d = 1$ with pivot points $A = 0$ and $B = 1/2$. In order to show the hardness result, we reduce QuadraticProgram-KKT problem to the local solution of LocalTripletLoss-Euclidean. The former has been shown to be CLS-complete [Fearnley et al., 2024]. Formally, we consider the following optimization problem

$$\min_{\boldsymbol{x} \in [0,1]^n} \boldsymbol{x}^\top \mathbf{Q} \boldsymbol{x} + \boldsymbol{b}^\top \boldsymbol{x}. \tag{1}$$

To show that QuadraticProgram-KKT reduces to LocalTripletLoss-Euclidean, we consider an arbitrary quadratic function of 3 variables $x, y, z \in [0, 1]$

$$q(x, y, z) := c_1 x^2 + c_2 y^2 + c_3 z^2 + c_4 xy + c_5 xz + c_6 yz + c_7 x + c_8 y + c_9 z. \tag{2}$$

For a LocalTripletLoss-Euclidean problem with points from $[0, 1]$, we set the margin $\alpha = 1$. Notice that for any triplet of points $(a_i, a_j, a_k) \in [0, 1]^3$, we have

$$(a_i - a_j)^2 - (a_i - a_k)^2 + 1 \geq 0.$$

Thus for any triplet constraint $(a_i, a_j, a_k)$ with weight $w$, the loss function is

$$\mathcal{L} = w \left( (a_i - a_j)^2 - (a_i - a_k)^2 + 1 \right).$$

For any triplet $(a_i, a_j, a_k)$, we denote triplet $(a_i, a_k, a_j)$ as its dual. Notice if $(a_i, a_j, a_k)$ has weight $w$ and the dual triplet $(a_i, a_k, a_j)$ has weight $w'$, we have the following objective function

$$\mathcal{L} = (w - w') \left( (a_i - a_j)^2 - (a_i - a_k)^2 \right) + w + w'.$$

Now consider a LocalTripletLoss-Euclidean instance on $(x, y, z) \in [0, 1]^3$. Triplets with weights $w_i$ are given as

Triplet $t_1 = (x, 0, y)$ with weight $w_1$: $\mathcal{L}_1 = w_1 \left( (x - 0)^2 - (x - y)^2 + 1 \right)$;

triplet $t_2 = (y, 0, x)$ with weight $w_2$: $\mathcal{L}_2 = w_2 \left( (y - 0)^2 - (y - x)^2 + 1 \right)$;

triplet $t_3 = (x, 0, z)$ with weight $w_3$: $\mathcal{L}_3 = w_3 \left( (x - 0)^2 - (x - z)^2 + 1 \right)$;

triplet $t_4 = (z, 0, x)$ with weight $w_4$: $\mathcal{L}_4 = w_4 \left( (z - 0)^2 - (z - x)^2 + 1 \right)$;

triplet $t_5 = (y, 0, z)$ with weight $w_5$: $\mathcal{L}_5 = w_5 \left( (y - 0)^2 - (y - z)^2 + 1 \right)$;

triplet $t_6 = (z, 0, y)$ with weight $w_6$: $\mathcal{L}_6 = w_6 \left( (z - 0)^2 - (z - y)^2 + 1 \right)$;

triplet $t_7 = (x, 0, \frac{1}{2})$ with weight $w_7$: $\mathcal{L}_7 = w_7 \left( x^2 - (x - \frac{1}{2})^2 + 1 \right)$;

triplet $t_8 = (y, 0, \frac{1}{2})$ with weight $w_8$: $\mathcal{L}_8 = w_8 \left( y^2 - (y - \frac{1}{2})^2 + 1 \right)$;

triplet $t_9 = (z, 0, \frac{1}{2})$ with weight $w_9$: $\mathcal{L}_9 = w_9 \left( z^2 - (z - \frac{1}{2})^2 + 1 \right)$;

triplet $t_{10} = (0, x, \frac{1}{2})$ with weight $w_{10}$: $\mathcal{L}_{10} = w_{10} \left( (0 - x)^2 - \frac{1}{4} + 1 \right)$;

triplet $t_{11} = (0, y, \frac{1}{2})$ with weight $w_{11}$: $\mathcal{L}_{11} = w_{11} \left( (0 - y)^2 - \frac{1}{4} + 1 \right)$;

triplet $t_{12} = (0, z, \frac{1}{2})$ with weight $w_{12}$: $\mathcal{L}_{12} = w_{12} \left( (0 - z)^2 - \frac{1}{4} + 1 \right)$.

The duals of above constraints are defined similarly with weight $w_i'$. Summing up all the objectives we get

$$
\begin{aligned}
\mathcal{L} = {} & (w_1 - w_1') \left( 2xy - y^2 \right) + (w_2 - w_2') \left( 2xy - x^2 \right) + (w_3 - w_3') \left( 2xz - z^2 \right) \\
& + (w_4 - w_4') \left( 2xz - x^2 \right) + (w_5 - w_5') \left( 2yz - z^2 \right) + (w_6 - w_6') \left( 2yz - y^2 \right) \\
& + (w_7 - w_7')x + (w_8 - w_8')y + (w_9 - w_9')z \\
& + (w_{10} - w_{10}')x^2 + (w_{11} - w_{11}')y^2 + (w_{12} - w_{12}')z^2 + C.
\end{aligned}
$$

We note that the last constant term $C$ wouldn't change the first-order stationary point of the above program. By setting

$$
\begin{aligned}
& w_1 - w_1' = -c_2; \\
& w_2 - w_2' = c_2 + \tfrac{c_4}{2}; \\
& w_3 - w_3' = -c_3 - \tfrac{c_6}{2}; \\
& w_4 - w_4' = c_3 + \tfrac{c_5}{2} + \tfrac{c_6}{2}; \\
& w_5 - w_5' = \tfrac{c_6}{2}; \\
& w_6 - w_6' = 0; \\
& w_7 - w_7' = c_7; \\
& w_8 - w_8' = c_8; \\
& w_9 - w_9' = c_9; \\
& w_{10} - w_{10}' = c_1 + c_2 + c_3 + \tfrac{c_4}{2} + \tfrac{c_5}{2} + \tfrac{c_6}{2}; \\
& w_{11} - w_{11}' = 0; \\
& w_{12} - w_{12}' = 0,
\end{aligned}
$$

we have $\mathcal{L} = q(x, y, z)$. This means that from any first-order stationary point $(x^*, y^*, z^*)$ of the objective $\mathcal{L}$, we have $[x^*, y^*, z^*]^\top \in [0, 1]^3$ is a KKT for the quadratic program defined in (2).

As shown above, $\mathcal{L}$ has the power to represent any quadratic polynomials $q(x, y, z)$. For the general form as in (1), one can group variables into groups of three $(v_i, v_j, v_k)$ and repeat the construction shown above. Since at most we have $O(n^2)$ interacting terms $v_i v_j$ where $v_i$ and $v_j$ are different variables, it requires $O(n^2)$ triplets to represent the quadratic program in (1). From the CLS-hardness of QuadraticProgram-KKT, we conclude that LocalTripletLoss-Euclidean with embedding dimension $d = 1$ is CLS-hard.

To extend above result to higher dimensions, we construct a reduction similar to the one-dimension case. Since $\|\boldsymbol{a}_i - \boldsymbol{a}_j\|_2^2 = \|\boldsymbol{a}_i\|_2^2 + \|\boldsymbol{a}_j\|_2^2 - 2\boldsymbol{a}_i^\top \boldsymbol{a}_j$, by using the same constraints and coefficients as the one-dimensional case and setting $\alpha = d$, we can construct a quadratic program of the form

$$\min_{\boldsymbol{x} \in [0,1]^{dn}} \boldsymbol{x}^\top \mathbf{Q}\boldsymbol{x} + \boldsymbol{b}^\top \boldsymbol{x}. \tag{3}$$

Where $\mathbf{Q}$ is a $dn \times dn$ matrix with $n \times n$ blocks, the $(i,j)^{th}$ block is of form $c_{ij} \cdot \mathbf{I}$. $\boldsymbol{b} \in \mathbb{R}^{dn}$ is a concatenation of $n$ vectors $(\boldsymbol{v}_1, \ldots, \boldsymbol{v}_n)$, where each $\boldsymbol{v}_i \in \mathbb{R}^d$ is of form $d_i \cdot \mathbf{1}$ . Now for any first-order stationary point $\boldsymbol{a}^* = (\boldsymbol{a}_1^*, \boldsymbol{a}_2^*, \ldots, \boldsymbol{a}_n^*) \in [0,1]^{dn}$, we consider the first coordinate for each $a_{i,1}^* = \boldsymbol{a}_i^* \cdot \boldsymbol{e}_1$, one can verify that $\boldsymbol{a}_1^* = (a_{1,1}^*, \ldots, a_{n,1}^*) \in [0,1]^d$ forms a KKT point of the following program

$$\min_{\boldsymbol{x} \in [0,1]^n} \boldsymbol{x}^\top \mathbf{C}\boldsymbol{x} + \boldsymbol{d}^\top \boldsymbol{x}.$$

Where $\mathbf{C} \in \mathbb{R}^{n \times n}$ and $c_{ij}$ is the coefficient for the $(i,j)^{th}$ block of $\mathbf{Q}$ defined in (3) and $d_i$ is the coefficient of the $i^{th}$ vector of $\boldsymbol{b}$ in (3). From the CLS-hardness of QUADRATICPROGRAM-KKT, we conclude that LOCALTRIPLETLOSS-EUCLIDEAN is CLS-hard. □

# E    Detailed experimental statistics

Table 1: Experimental statistics for the hard instances $H_1$ to $H_{15}$ used in Section 5. The second and the third columns show the number of vertices and edges in the original LOCALMAXCUT instance. The next 6 columns show the number of vertices and constraints in the reduced LOCALBETWEENNESS-EUCLIDEAN-1D, LOCALCONTRASTIVE-EUCLIDEAN-1D and LOCALCONTRASTIVE-TREE instances. The last column shows the number of iterations needed to reach a local optimum.

| Instance | MaxCut | | Betweenness-1D | | Contrastive-1D | | Contrastive-Tree | | #iterations |
|---|---|---|---|---|---|---|---|---|---|
| | $n$ | $m$ | $n$ | $m$ | $n$ | $m$ | $n$ | $m$ | |
| $H_1$ | 37 | 45 | 38 | 45 | 40 | 121 | 41 | 204 | 63 |
| $H_2$ | 65 | 81 | 66 | 81 | 68 | 213 | 69 | 360 | 167 |
| $H_3$ | 93 | 117 | 94 | 117 | 96 | 305 | 97 | 516 | 375 |
| $H_4$ | 121 | 153 | 122 | 153 | 124 | 397 | 125 | 672 | 791 |
| $H_5$ | 149 | 189 | 150 | 189 | 152 | 489 | 153 | 828 | 1623 |
| $H_6$ | 177 | 225 | 178 | 225 | 180 | 581 | 181 | 984 | 3287 |
| $H_7$ | 205 | 261 | 206 | 261 | 208 | 673 | 209 | 1140 | 6615 |
| $H_8$ | 233 | 297 | 234 | 297 | 236 | 765 | 237 | 1296 | 13,271 |
| $H_9$ | 261 | 333 | 262 | 333 | 264 | 857 | 265 | 1452 | 26,583 |
| $H_{10}$ | 289 | 369 | 290 | 369 | 292 | 949 | 293 | 1608 | 53,207 |
| $H_{11}$ | 317 | 405 | 318 | 405 | 320 | 1041 | 321 | 1764 | 106,455 |
| $H_{12}$ | 345 | 441 | 346 | 441 | 348 | 1133 | 349 | 1920 | 212,951 |
| $H_{13}$ | 373 | 477 | 374 | 477 | 376 | 1225 | 377 | 2076 | 425,943 |
| $H_{14}$ | 401 | 513 | 402 | 513 | 404 | 1317 | 405 | 2232 | 851,927 |
| $H_{15}$ | 429 | 549 | 430 | 549 | 432 | 1409 | 433 | 2388 | 1,703,895 |

