# OpenReview forum: "The Complexity of Finding Local Optima in Contrastive Learning"
_NeurIPS.cc/2025/Conference — NeurIPS 2025 poster_

### Official Review · Reviewer_dqwU · 2025-06-30

**Clarity:** 3
**Significance:** 2
**Originality:** 3
**Rating:** 4
**Confidence:** 2

**Summary:**

This paper studies contrastive learning, where the input consists of a set of triplets $(x_i, y_i^+,z_i^- )$. This indicates $x_i$ is more similar to $y_i^+$ than to $z_i^-$. Each triplet is associated with weight $w_i$. The objective l is to learn an embedding such that the embedding of $x_i$ is closer to $y_i$ than to $z_i$ . The paper provides the computational complexity of finding locally optimal solutions with three loss formulations for both discrete and continuous settings. Its main contribution is providing proof that optimizing these contrastive loss functions is either PLS-hard or CLS-hard.

**Questions:**

- Could you clarify the difference between the continuous and discrete settings? I assume the distinction is whether the inputs  $(x_i, y_i^+,z_i^- )$ take values in a continuous or discrete space. But loss is optimized over the space of embedding parameters $\theta$. $\theta$ can be continuous even if the   $(x_i, y_i^+,z_i^- )$ are discrete?

- What does the weight $w_i$ represent in practice? How is the importance of a triplet determined?

- I am interested to know if you can provide a practical application of the setting.
- What would the complexity and suboptimality in an online (dynamic) optimization setting, where the model is updated based on the arrival of each new triplet?
- Reinforcement Learning from Human Feedback (RLHF) and Direct Preference Optimization (DPO) also consider learning from pairwise preference, where users indicate that they prefer $y$ to $z$. Can the theoretical analysis in this paper be extended to cover such settings?

**Ethical Concerns:**

["NO or VERY MINOR ethics concerns only"]

**Final Justification:**

The authors have addressed my question about the paper. As I stated prior to the rebuttal, this is primarily a theoretical work. While I agree with Reviewers XHnj and VGp6 that additional empirical analysis could further strengthen the paper by demonstrating its practical applicability, I believe the theoretical contributions are well-presented and solid. So, I will maintain my positive rating.

**Limitations:**

I do not see any immediate potential negative societal impact.

**Quality:**

2

**Strengths And Weaknesses:**

Strengths:
Comprehensive Theoretical study.

Weakness:
Lack of empirical validation. Currently the result show the number of iterations increase exponentially with the number of vertices. How suboptimal the result would be if it is run using SGD for a finite number of epcohs, which is common in learning embedding.

---

> ### Author Rebuttal · Authors · 2025-07-30
>
> We thank the reviewer for their time and positive feedbacks.
>
> > "Lack of empirical validation."
>
> The main focus of this paper deals with worst-case runtime analysis of local search methods in various contrastive settings and the purpose of our experiments is only to validate our theoretical findings. As we point out in the Conclusion section (line 334 - 338), we deem other theoretical frameworks such as average-case analysis, smoothed analysis, beyond worst-case analysis, and the corresponding empirical validations interesting but go well beyond the scope of our work. For more details regarding the significance of worst-case analysis along with other theoretical models that captures more practical settings, we refer the reviewer to our discussion with Reviewer VGp6 and Reviewer XHnj.
>
> > "Currently the result show the number of iterations increase exponentially with the number of vertices. How suboptimal the result would be if it is run using SGD for a finite number of epcohs, which is common in learning embedding."
>
> The objective function we consider in the experiment is the discrete objective function $\sum_{i = 1}^m w_i \cdot \mathbf{1}_{(x_i, y_i, z_i)}$ (see our response below regarding the differences between discrete and continuous settings). Due to its discrete nature, it would not be applicable to perform gradient descent on this problem. The local search method in our experiment is natural to the discrete setting and, even the convergence rate is exponential, is guaranteed to converge to a locally optimal point.
>
> > "Could you clarify the difference between the continuous and discrete settings? I assume the distinction is whether the inputs $(x_i, y_i^+, z_i^-)$ take values in a continuous or discrete space. But loss is optimized over the space of embedding parameters $\theta$. $\theta$ can be continuous even if the $(x_i, y_i^+, z_i^-)$ are discrete?"
>
> The main distinction between the continuous and discrete settings is whether the objective function is a continuous function. For discrete settings, the typical objective function we consider is $\sum_{i = 1}^m w_i \cdot \mathbf{1}_{(x_i,y_i, z_i)}$ which is a weighted sum of indicators of whether triplet $(x_i, y_i, z_i)$ is satisfied (which is discontinuous). For continuous settings, the objective we consider is the triplet loss function defined in line 186. Notice that this objective function is continuous and we can perform gradient descent to continuous settings.
>
> Regarding the solution space, (i.e., the "space of embedding parameters $\theta$"), it is a continuous space (either $R^d$ or $[0, 1]^d$) in all Euclidean embedding settings (regardless of whether the objective function is continuous or discrete), and is a discrete space in the tree embedding setting (LocalContrastive-Tree).
>
> > "What does the weight $w_i$ represent in practice? How is the importance of a triplet determined?"
>
> The weights $w_i$ serve as a way to determine the importance of given triplets and are usually provided as the input of the problem, just like how the weights of edges are provided in a graph. One way to formulate the weight is that if we consider triplet $(x, y^+, z^-)$ as the constraint that the embedding needs to satisfy, where $w_i$ could be viewed as the cost if such constraint is not satisfied.
>
> > "I am interested to know if you can provide a practical application of the setting."
>
> As we discussed in the Introduction (line 36-43), many interesting and widely-used applications in our setting include nearest-neighbor, recommendation, ranking, and crowdsourcing. As for our theoretical findings, we would like to emphasize that our paper focuses on providing the first computational hardness results in such settings and could serve as a stepping stone for interesting future works that investigate practical behavior over real-world datasets. We refer the reviewer to our discussion with Reviewer VGp6 and Reviewer XHnj for the significance of worst-case analysis and other theoretical frameworks that capture more practical settings.
>
> > "What would the complexity and suboptimality in an online (dynamic) optimization setting, where the model is updated based on the arrival of each new triplet?"
>
> We address that all our setting assume the algorithms have full access to the input set of triplets and their corresponding weights. Accessing the triplets in an online setting is at least as hard as the full information setting so all our hardness results extend to the online setting.
>
> > "Reinforcement Learning from Human Feedback (RLHF) and Direct Preference Optimization (DPO) also consider learning from pairwise preference, where users indicate that they prefer $y$ to $z$. Can the theoretical analysis in this paper be extended to cover such settings?"
>
> We can formulate RLHF and DPO problems in a contrastive setting such that the model queries are considered as anchor points ($x$), the preferred choices are treated as positive points ($y^+$), and the less preferred choices are viewed as negative points ($z^-$). If the objective functions in this formulation match the settings we study in this work (Contrastive-Euclidean, Contrastive-Tree, Betweenness-Euclidean, or TripletLoss), then our hardness results apply to those problems.
>
> We would like to thank the reviewer again for their valuable time and helpful comments. We hope our response addresses their concerns. We would really appreciate if the reviewer would reconsider their evaluation of our work.
>
> Best regards,
>
> The authors.

---

> > ### Comment · Reviewer_dqwU · 2025-08-05
> >
> > Thank you for the reply. I like the theoretical contribution of this paper. I will keep my score, leaning toward accept.

---

### Official Review · Reviewer_XHnj · 2025-07-02

**Clarity:** 3
**Significance:** 2
**Originality:** 3
**Rating:** 2
**Confidence:** 3

**Summary:**

The paper worked on the complexity of finding local minimum of contrastive learning loss. The main idea is to reduce a polynomial local search or continuous local search problem to the triple minimization/maximization problem, then use the existence theory of PLS and CLS to show the corresponding complexities. The bridge is the LocalMaxCut problem, which was proved to be PLS-hard. The author worked on several variants/cases in 1D and $\geq 2$-D of the contrastive loss. i.e LocalConstrastive-Euclidean problem, Local Constrastive-Tree problem, LocalBetweenness-Euclidean problem in both discrete case and continuous case. The main ideas in all the proofs are to add some certain triples that have height weights to show the equivalence between two problems.

**Questions:**

Based on two points stated in the weaknesses, my decision is to reject, although I am not familiar with most of the literature.

**Ethical Concerns:**

["NO or VERY MINOR ethics concerns only"]

**Final Justification:**

I keep my score for definite rejection, since I think the paper has very little contributions and impact to the community. The conference should prioritize the place for works with practical benefit.

**Limitations:**

Yes, it is theoretical work.

**Quality:**

2

**Strengths And Weaknesses:**

Strengths:

1. I think the author make an attempt to introduce the problem and all related works, since I believe they are not familiar with AI conferences.
2. The presentation is well-structured and clear.
3. I believe that the results have some valid contribution to the theory of complexity of contrastive learning loss.

Weaknesses:

1. As a practitioner, I do not find any practical benefit from this work when working with contrastive learning.
2. I am not familiar with most of the literature mentioned in the paper. However, I find the proofs are based on combinatorial structures, but all these structures are quite simple. Thus, I do not appreciate the novelty as well as the difficulty of presented proofs.
3. No applications.

---

> ### Author Rebuttal · Authors · 2025-07-30
>
> We thank the reviewer for their time and the reviews they provide.
>
> > "As a practitioner, I do not find any practical benefit from this work when working with contrastive learning."
>
> And also
>
> > "No applications."
>
> This is a Theory Track paper dealing with fundamental optimization questions. The theory of computation, celebrated by multiple Turing awards, Godel prizes, and various other mathematical/Computer Science prizes, discusses the worst-case complexity of finding solutions to optimization problems, and we believe our work showing worst-case results is a stepping stone towards understanding the performance on real-world instances. Through our reductions, we highlight the hard-instances of our problems, and typically, worst-case results have yielded important complexity classes, progressed the community's understanding on computation and computability. From a practical point of view, there are theoretical frameworks such as average-case analysis, smoothed analysis, beyond-worst-case analysis (see Roughgarden's book on beyond-worst case). Before moving towards such beyond-worst-analysis theoretical guarantees, one needs to firstly understand the worst-case. Our work serves this purpose and should be considered as stand-alone work.
>
> > "I find the proofs are based on combinatorial structures, but all these structures are quite simple. Thus, I do not appreciate the novelty as well as the difficulty of presented proofs."
>
> We deem our work highly technical and that it carries significant theoretical novelties. Computational hardness reductions typically involve creating gadgets, such as extra variables, structured functions and geometric configurations etc., which are fundamental building blocks for **all** reductions in computational complexity. Our work follows the convention of typical reductions which operate on purely combinatorial objects such as graphs, trees, circuits, permutations, and embeddings. While some of the gadgets might look easy at the first glance, coming up with such gadgets require much more efforts as we need to preserve the local optimality from our settings to problems we reduce to.
>
> Furthermore, prior to our work, the complexity of finding local optimum in our settings remained open and is a fundamental question in optimization. The PLS/CLS-hardness results in this paper provide strong evidence that the problems we consider are computationally hard and it is unlikely for polynomial-time algorithm to find local solutions. This is the first work that establishes the computational complexity results for contrastive learning and we believe it carries fundamental theoretical contributions to broader machine learning communities.
>
> Below we provide summaries of the challenges and technical contributions of our reductions.
>
> * **Discrete Contrastive and Betweenness.** When the dimension is one (on a real line), it is not hard to simulate a cut of a graph through discrete contrastive triplets since all vertices of the graph either fall into the left side or the right side of some reference vertex $X$. However, this problem becomes **significantly harder** in high dimensional space. One of the main challenges is that the notion of left side and right side of reference vertex $X$ becomes unclear in high dimensional spaces. In fact, if we construct the same triplets as in the 1D case, all the vertices will be located on different circles centered at $X$ and we would not be able to simulate a cut out of such geometrical configuration of vertices. The key novelty of the high dimensional reduction comes from the Isosceles gadget and the the Equilateral gadget (line 494 - 499), which allow us to simulate a cut of the graph through the relative position between the vertices and a regular simplex enforced by the gadgets. As a result, the triplets we construct in high dimensional case are drastically different from the ones in 1D case and exhibit significant technical insights.
>
> * **Discrete Tree embeddings.** The main challenge for tree embeddings is that unlike the Euclidean embeddings where we can leverage on the properties of the geometric configuration of vertices, tree embeddings are purely combinatorial and lack inherent geometric properties. To bypass this difficulty, we introduce several gadgets that effectively enforce all the vertices to lie on one of the two subtrees. In this way we can simulate a cut of the graph through checking which subtree each vertex lies in. We believe that the gadgets implemented in the tree reduction introduce strong theoretic contributions and could be of independent interest for other hardness reductions involving tree structures.
>
> * **Continuous contrastive learning.** The principle observation for the continuous contrastive learning reduction is to identify the significance of pivot points. As we emphasized in line 189-193, without the introduction of pivot points, the learning task would become trivial as there would be trivial first-order stationary point where all points overlap with each other. Another technical obstacle for this reduction is the coupling between the triplet constructed due to shared variables (line 288-295). We manage to overcome this barrier and reconstruct any quadratic program with three variables through a careful design of triplets and corresponding weights. Finally, we acquire the CLS-hardness result by extending the reduction to high dimensional quadratic programs.
>
> We thank the reviewer again for the time they spent on providing feedbacks. We hope our comments address their concerns. We would really appreciate if they would reconsider their evaluation of our work.
>
> Best regards,
>
> The authors.

---

> > ### Comment · Reviewer_XHnj · 2025-08-05
> > **Reply to the rebuttal**
> >
> > I would like to thank the authors for their replies. In my opinion, this is AI conference, so the paper needs to show practical benefit and applications in AI. I do not object if the paper is submitted to other conferences of Computer Science.
> >
> > I still think the combinatorial/geometrical  structures in the proofs are simple. So I do think the proofs is mathematically simple.

---

> ### Author Response · Authors · 2025-08-05
> **Further clarification**
>
> We thank the reviewer for taking the time to read our response and raising additional concerns. We would like to take this opportunity to provide further clarification.
>
> > Suitability of the venue
>
> As outlined in our rebuttal, we believe the practical implications of our work are twofold: i) This is the first rigorous study of the computational complexity of finding local stationary point in contrastive learning, which is an area of significant interest in both theoretical and applied research; and ii) our results may server as a stepping stone for further investigations into other theoretical frameworks that capture more applied settings. Regarding the suitability of NeurIPS, we would like to emphasis that worst-case analysis and computational hardness results are well represented in machine learning conferences, including ICML, ICLR, NeurIPS, AAAI, and COLT. For reference, we provide a list of papers addressing computational hardness that have been published in these venues [1-8]. Moreover, since the scope of our paper focuses on contrastive learning, a topic of broad relevance and active research in the machine learning community, we believe NeurIPS is an appropriate venue for our work.
>
> > Proof structure
>
> Regarding the structure of the proof, we would like to reiterate that while the mathematical tools employed in this work are relatively elementary, the overall construction and analysis are far from simple. In our view, although the proofs may be straightforward to follow, the process of designing gadgets that simultaneously enforce certain combinatorial structures, such as a cut in a graph, while preserving local optimality is highly nontrivial and technically involved.
>
> Best regards,
>
> The authors.
>
> ________
>
> References:
>
> [1] Constantinos Daskalakis, Noah Golowich, and Kaiqing Zhang. The complexity of markov equilibrium in stochastic
> games. COLT 2023.
>
> [2] Argyrios Deligkas, John Fearnley, Alexandros Hollender, and Themistoklis Melissourgos. Tight inapproximability for
> graphical games. AAAI 2023.
>
> [3] Edith Elkind, Leslie Ann Goldberg, Paul Goldberg, and Michael Wooldridge. Computational complexity of weighted
> threshold games. AAAI 2007.
>
> [4] Edith Elkind, Leslie Ann Goldberg, Paul Goldberg, and Michael Wooldridge. On the dimensionality of voting games. AAAI 2008.
>
> [5] Edith Elkind, Abheek Ghosh, and Paul Goldberg. Complexity of deliberative coalition formation. AAAI 2022.
>
> [6] Alexandros Hollender and Emmanouil Zampetakis. The computational complexity of finding stationary points in
> non-convex optimization. COLT 2023.
>
> [7] Alexandros Hollender, Gilbert Maystre, and Sai Ganesh Nagarajan. The complexity of two-team polymatrix games
> with independent adversaries. ICLR 2025.
>
> [8] Andreas Kontogiannis, Vasilis Pollatos, Sotiris Kanellopoulos, Panayotis Mertikopoulos, Aris Pagourtzis, and Ioannis
> Panageas. The computational complexity of finding second-order stationary points. ICML 2024.

---

### Official Review · Reviewer_VGp6 · 2025-07-02

**Clarity:** 2
**Significance:** 2
**Originality:** 3
**Rating:** 4
**Confidence:** 2

**Summary:**

This paper investigates the computational complexity of finding local optima in contrastive learning, proving PLS-hardness for discrete objectives and CLS-hardness for continuous objectives, indicating that no polynomial-time algorithm can find local optima unless complexity classes collapse. It provides theoretical foundations for understanding the optimization challenges in contrastive learning and suggests future research directions, including average-case complexity and the quality of local solutions.

**Questions:**

1. Could the authors please clarify how worst-case scenario analysis for local optima contributes to either optimization efficiency or convergence reliability, and whether there are any existing mitigation strategies for such scenarios?

2. The experimental evidence (Figure 5) relies on specific worst-case constructions inherited from LocalMaxCut. Is there empirical evidence or theoretical argument about how these scenarios relate to typical or randomly generated instances of contrastive learning? Do the exponential local search behaviors manifest on real datasets?

**Ethical Concerns:**

["NO or VERY MINOR ethics concerns only"]

**Final Justification:**

The theoretical framework presented is sound, yet the work would benefit considerably from more comprehensive empirical analysis to substantiate its practical applicability.

**Limitations:**

The paper does not include extensive empirical experiments to validate the theoretical results. While the constructed hard instances demonstrate the potential for exponential time complexity, the paper does not provide a broad empirical study across different datasets and initialization methods.

**Paper Formatting Concerns:**

N.A.

**Quality:**

2

**Strengths And Weaknesses:**

Strengths：
1. By focusing on the complexity of local optima (rather than just global optima), the paper connects empirical observations of slow or stuck convergence in contrastive learning to formal complexity-theoretic barriers.
2. The paper delivers a comprehensive set of hardness results: PLS-hardness for maximization of contrastive triplets and betweenness objectives in both R^d and on trees, and CLS-hardness for continuous triplet loss minimization. These are rigorously demonstrated via reductions from classic hard problems (LocalMaxCut, QuadraticProgram-KKT).

Weaknesses：
1. There is little discussion or exploration of practical scenarios or real datasets in experiments --- leaving open the question of whether these hardness results are representative of typical training behavior in common machine learning settings beyond theoretical constructions.
2. The paper focuses on the complexity in the worst case, but does not analyze the behavior in the average case, which makes it difficult to fully understand the complexity of local optima in contrastive learning.
3. There are no corresponding methods or experiments proposed to address the complexity caused by the worst-case scenarios.

---

> ### Author Rebuttal · Authors · 2025-07-30
>
> We thank the reviewer for their valuable time and suggestions.
>
> > "There is little discussion or exploration of practical scenarios or real datasets in experiments --- leaving open the question of whether these hardness results are representative of typical training behavior in common machine learning settings beyond theoretical constructions."
>
> And also
>
> >"The paper focuses on the complexity in the worst case, but does not analyze the behavior in the average case, which makes it difficult to fully understand the complexity of local optima in contrastive learning."
>
> The paper addresses the worst-case runtime complexity of various contrastive settings via local search methods, including gradient descent, which was open prior to our work. Finding local optima is a fundamental question and we present hardness results for them. As such, all results are in terms of worst-case analysis, similar to proofs of NP-hardness for worst-case instances of combinatorial problems.
>
> As mentioned in the Conclusion (line 334 - 338), real datasets are expected to exhibit non-worst-case behaviors. Moreover, we suggested studying the average-case behavior through other theoretical frameworks such as smoothed analysis in our settings as one of the future directions. It is worth noting that for a lot of classical problems like local Maximum-cut or computing Nash equilibria in general-sum games, the corresponding smoothed analysis results [1, 2] are proven years later than the computational complexity results and they often require new theoretical insights about the problems. We believe this submission is a stand-alone work and, apart from its own theoretical novelty, it serves as a stepping stone for future studies.
>
> Finally, we would like emphasize that this paper is submitted under the Theory primary area, as it mainly focuses on a fundamental question instead of studying practical behaviors over real-world datasets. The key takeaway is that finding local minima requires exponential time in the worst case.
>
> > "There are no corresponding methods or experiments proposed to address the complexity caused by the worst-case scenarios."
>
> And also
>
> > "Could the authors please clarify how worst-case scenario analysis for local optima contributes to either optimization efficiency or convergence reliability, and whether there are any existing mitigation strategies for such scenarios?"
>
> The theory of computation deals first with worst-case complexity, and this was open for finding local minima in all our settings. Our goal was to answer this open question and we did this for the worst-case analysis. Similar to the theory of NP-hardness that provides intractability evidence for finding global optima, our work provides strong intractability evidence for finding local optima, which is more challenging. In fact, our hardness results preclude the existence of any polynomial-time algorithm that finds a locally optimum in the settings we study unless there is a collapse of complexity classes (PLS $\subseteq$ P or CLS $\subseteq$ P).
>
> We would like to highlight that our goal was not to bypass hardness in real-life scenarios, similar to how proofs of NP- or PPAD-hardness discuss the underlying complexity of the problem (e.g., 3-SAT, Nash Equilibria etc) and not real-world performance of standard heuristics. Methods that avoid worst-case scenarios in practice could be an interesting future direction to investigate. As we noted in the Conclusion section, one possible approach is to apply small random perturbations and study its smoothed complexity. It is noteworthy that smoothed analysis is problem-specific, existing polynomial-time result for PLS/CLS-complete problems [3] does not directly extend to the contrastive settings that we study.
>
> > "The experimental evidence (Figure 5) relies on specific worst-case constructions inherited from LocalMaxCut. Is there empirical evidence or theoretical argument about how these scenarios relate to typical or randomly generated instances of contrastive learning? Do the exponential local search behaviors manifest on real datasets?"
>
> As we discussed above, randomly generated instances (or instances under random perturbation) are interesting future direction we mentioned in Conclusion, but the scope of this submission focuses on worst-case complexity, which is a fundamental theoretical question on its own.
>
> We thank the reviewer again for their time and suggestions. We hope our comments address their concerns about this paper. We would really appreciate if the reviewer would reconsider their evaluation of our work.
>
> Best regards,
>
> The authors.
>
> --------
> References:
>
> [1] Michael Etscheid and Heiko R¨oglin. “Smoothed analysis of local search for the maximum-cut problem”. In: Proceedings of the Twenty-Fifth Annual ACM-SIAM Symposium on Discrete Algorithms. SODA ’14. Portland, Oregon: Society for Industrial and Applied Mathematics, 2014, pp. 882–889.
>
> [2] Shant Boodaghians et al. “Smoothed complexity of 2-player Nash equilibria”. In: 2020 IEEE 61st Annual Symposium on Foundations of Computer Science (FOCS). IEEE. 2020, pp. 271–282.
>
> [3] Pavel Huáček and Eylon Yogev. “Hardness of Continuous Local Search: Query Complexity and Cryptographic Lower Bounds”. In: Proceedings of the 2017 Annual ACM-SIAM Symposium on Discrete Algorithms (SODA), pp. 1352–1371.

---

> > ### Comment · Reviewer_VGp6 · 2025-08-06
> >
> > While I appreciate the authors' response to my initial concerns, I remain unconvinced that the core issues regarding practical utility and computational efficiency have been adequately addressed. The theoretical contributions presented are noteworthy; however, deep learning is fundamentally a practically-oriented field where empirical validation carries significant weight. Given these, I strongly encourage the authors to provide more extensive empirical support for their theoretical insights. I am willing to consider adjusting my evaluation, provided the authors can better bridge the theory-practice divide in the final revision.

---

> ### Author Response · Authors · 2025-08-07
> **Further justification**
>
> We thank the reviewer for their insightful feedback. Below we further address the concerns raised by the reviewer.
>
> - First, we would like to emphasize that, although highly theoretical, worst-case analysis and computational hardness results are well represented in machine learning conferences, including ICML, ICLR, NeurIPS, AAAI, and COLT. For reference, we provide a list of papers addressing computational hardness that have been published in these venues [1 - 8]. Notably, **all of these works** focus exclusively on their theoretical findings and do not include any experimental validations.
>
> - That being said, we fully agree with the reviewer that strong theoretical works should ideally offer practical insights as well. We provide additional preliminary experimental results in the table below. To simulate more practical scenarios, we apply uniformly random perturbations to the weights of the hard instance constructed in Section 5 (line 303 - 327). When the magnitude perturbation is small $(\epsilon = \pm 0.3)$, local search methods still require exponential time to stabilize. This **further strengthens our theoretical findings under slight perturbations**. In contrast, with large perturbations $(\epsilon = \pm 1)$, the construction resembles more randomly generated triplets, and empirical results indicate that local search methods are quick and converge in linear time. We would like to emphasize that this contrast has already been discussed in the Conclusion section (line 335 - 338) where we acknowledge that many similar problems admit efficient algorithms beyond the worst-case setting. Furthermore, we reiterate that providing rigorous smoothed analysis is beyond the scope of this work; however, we view it as a significant direction for future.
>
> | **Number of Vertices** | **Number of Moves (Mean ± Std, ε = 0.3)** | **Number of Moves (Mean ± Std, ε = 1.0)** |
> |------------------------|--------------------------------------------|--------------------------------------------|
> | 40                     | 62.62 ± 2.79                               | 5.82 ± 6.96                                 |
> | 68                     | 162.19 ± 24.36                             | 5.59 ± 6.16                                 |
> | 96                     | 370.95 ± 28.56                             | 5.51 ± 6.78                                 |
> | 124                    | 783.41 ± 75.52                             | 6.60 ± 6.35                                 |
> | 152                    | 1608.38 ± 145.47                           | 5.75 ± 6.06                                 |
> | 180                    | 3167.16 ± 587.95                           | 6.33 ± 6.12                                 |
> | 208                    | 6421.07 ± 1102.95                          | 6.59 ± 6.40                                 |
> | 236                    | 12664.28 ± 2661.35                         | 7.37 ± 6.35                                 |
> | 264                    | 26320.04 ± 2616.42                         | 9.36 ± 8.42                                 |
>
> - Regarding the practical insights of our results, we would like to address that the hard instances constructed in our reduction require certain anchor vertices $X$ to appear in a polynomial number of triplets ($poly(m, n)$). We believe this property is one of the fundamental components in constructing hard instances, as it ensures that altering the permutation of an anchor vertex $X$ could potentially influence the permutations of all other vertices associated with the same triplets as $X$. In other words, our reduction suggests that **the computational hardness of contrastive learning is closely tied to the degree of the dependency hypergraph of the triplets** (i.e., the number of triplets in which a given vertex appears).
>
> We thank the reviewer again for their valuable time and feedback. In the revised version, we will more extensively emphasize how the theoretical findings of our work (i) contribute to a better understanding of practical performance through perturbed instances, and (ii) highlight necessary conditions on the triplets that make contrastive learning computationally hard.

---

> ### Author Response · Authors · 2025-08-07
> **Supplemental references**
>
> ________
> References:
>
> [1] Constantinos Daskalakis, Noah Golowich, and Kaiqing Zhang. The complexity of markov equilibrium in stochastic games. COLT 2023.
>
> [2] Argyrios Deligkas, John Fearnley, Alexandros Hollender, and Themistoklis Melissourgos. Tight inapproximability for graphical games. AAAI 2023.
>
> [3] Edith Elkind, Leslie Ann Goldberg, Paul Goldberg, and Michael Wooldridge. Computational complexity of weighted threshold games. AAAI 2007.
>
> [4] Edith Elkind, Leslie Ann Goldberg, Paul Goldberg, and Michael Wooldridge. On the dimensionality of voting games. AAAI 2008.
>
> [5] Edith Elkind, Abheek Ghosh, and Paul Goldberg. Complexity of deliberative coalition formation. AAAI 2022.
>
> [6] Alexandros Hollender and Emmanouil Zampetakis. The computational complexity of finding stationary points in non-convex optimization. COLT 2023.
>
> [7] Alexandros Hollender, Gilbert Maystre, and Sai Ganesh Nagarajan. The complexity of two-team polymatrix games with independent adversaries. ICLR 2025.
>
> [8] Andreas Kontogiannis, Vasilis Pollatos, Sotiris Kanellopoulos, Panayotis Mertikopoulos, Aris Pagourtzis, and Ioannis Panageas. The computational complexity of finding second-order stationary points. ICML 2024.

---

### Official Review · Reviewer_Fqic · 2025-07-03

**Clarity:** 3
**Significance:** 3
**Originality:** 3
**Rating:** 5
**Confidence:** 4

**Summary:**

The paper studies the computational complexity of finding local optima in contrastive learning problems, a gap previously unexplored compared to global optimization.
They prove that finding local optima for both discrete and continuous contrastive objectives is PLS-hard or CLS-hard, respectively, even in low-dimensional settings.
This establishes that no polynomial-time algorithm can guarantee finding such optima unless major complexity class collapses (PLS ⊆ P or CLS ⊆ P) occur.
The authors establish PLS- and CLS-hardness of contrastive learning objectives by reducing from well-known hard problems such as LocalMaxCut (for discrete settings) and QuadraticProgram-KKT (for continuous settings). These reductions are designed to preserve local optimality, using specialized constraint gadgets that encode the structure of the original problems within the contrastive framework.

The authors used a synthetic graph from Monien and Tscheuschner (2010), known to be hard for local search, and transformed it into instances of contrastive learning problems via their reductions.
Their goal was to test whether local search behaves similarly across problems; the result showed identical exponential-time convergence, validating that worst-case hardness transfers empirically.

**Questions:**

1. In lines 118--121, you state that the PLS/CLS-hardness results extend to $\varepsilon$-approximate local optima. Could you clarify for what values of $\varepsilon = \varepsilon(n, m)$---where $n = |V|$ (number of data points) and $m = |C|$ (number of triplet constraints)---the reductions remain valid?

2.  Can you explicitly define the geometric configuration of points X, Y, Z in the 1D embedding?

3. In lines 223--228, the proof shows that violating a boundary constraint results in a net loss, but it assumes that the total loss is upper bounded by $\sum w_{uv}$. Why this upper bound always dominates the gain from violating any single boundary constraint? Including a concrete inequality to justify this would help formalize the argument and strengthen the correctness of the reduction.

4. Theorem 4.1, It would be very helpful if you could list the 12 triplets used to encode a quadratic over $x, y, z$, and share a simple example (either symbolic or numeric) of the corresponding weights. This would really clarify how cross-terms like $xy$ are handled in the reduction.

**Ethical Concerns:**

["NO or VERY MINOR ethics concerns only"]

**Limitations:**

Yes

**Quality:**

3

**Strengths And Weaknesses:**

Strengths
- Novelty: First formal study of the complexity of local optima in contrastive learning.
Technical depth: Strong PLS and CLS reductions from canonical problems (LocalMaxCut, QuadraticProgram-KKT).
- Generality: Hardness holds even in low dimensions (e.g., $d=1$), and across Euclidean, tree, and loss-based settings.
- Theoretical rigor: Reductions are clean, constructions are well-motivated, and results are sharp.

Weaknesses
- Clarity: Some sections (especially CLS reductions) are dense; The paper assumes background in PLS/CLS; adding more intuition and context would help make it easier to follow for the broader ML community.
- Empirical scope: Experiments are limited to synthetic worst-case instances. This is acceptable since the paper was submitted under Theory as the primary area, though broader empirical validation could strengthen future versions.

---

> ### Author Rebuttal · Authors · 2025-07-30
>
> We thank the reviewer for their suggestions and positive feedbacks.
>
> > "Some sections are dense; The paper assumes background in PLS/CLS; adding more intuition and context would help make it easier to follow for the broader ML community."
>
> Due to the space constraint, we are unable to include full details in the main paper. For the CLS reduction, the intuition is provided in section 4 and the formal mathematical proof is presented in section D of the Appendix. In the proof, we firstly present a reduction to quadratic programs with three variables, which we believe is more intuitive, and then we extend the result to higher dimension settings. We also refer the reviewer to our discussion with Reviewer XHnj regarding the intuition and technicality of our reductions. If space allows, we will include more background information regarding PLS/CLS in the revised version.
>
> > "Experiments are limited to synthetic worst-case instances."
>
> The main focus of this paper deals with worst-case runtime analysis of local search methods in various contrastive settings and the purpose of our experiments is only to validate our theoretical findings. As we point out in the Conclusion section (line 334 - 338), we deem other theoretical frameworks such as average-case analysis, smoothed analysis, beyond worst-case analysis, and the corresponding empirical validations interesting but go well beyond the scope of our work. For more details regarding the significance of worst-case analysis along with other theoretical models that captures more practical settings, we refer the reviewer to our discussion with Reviewer VGp6 and Reviewer XHnj.
>
> > "Could you clarify for what values of $\epsilon = \epsilon(n, m)$—where $n = |V |$ (number of data points) and $m = |C|$ (number of triplet constraints)—the reductions remain valid?"
>
> Thanks for asking for clarifications on the accuracy parameter. Firstly, our PLS-hardness results hold for embeddings with discrete objectives ($\sum_{i = 1}^m w_i \cdot \mathbf{1}_{(x_i, y_i, z_i)}$),  and due to the discrete nature of the objective functions, approximating with small additive error $\epsilon$ is equivalent to computing an exact local optimum. Regarding our CLS-hardness results for the triplet loss objective, [1] showed $2^{\Omega(n)}$ query complexity in black-box model and further provided strong evidence that CLS is hard for polynomial-time algorithms when $\epsilon$ is inverse exponential ($1/exp(m, n)).$ Thus our results indicate that it is highly unlikely to find an $\epsilon$-approximate local optimal embedding in polynomial time when the error is inverse exponential.
>
> > "Can you explicitly define the geometric configuration of points $X$, $Y$, $Z$ in the 1D embedding?"
>
> Thanks for asking for clarifications on the geometric configuration. Since the question is a bit unclear to us, we address potential concerns in three different ways:
>
> 1. Regrading the triplet notation $(X, Y, Z)$, the formal definition can be found at the top of page 5, where $X$ is required to be closer to $Y$ than to $Z$ in the embedding space.
>
> 2. For the relative positions of the special vertices $X$, $Y$, $Z$ in any local optimal embedding, this is stated in lines 220-221: "either the order $X < Y < Z$ or $Z < Y < X$, with distances $|X - Y| \geq |Y - Z|$"; we also illustrated this in Figure 2.
>
> 3. Finally, to clarify "how the specific triplet constraints listed in the proof enforce this configuration", we provide the following case analysis covering all $6$ possible orderings of $X, Y, Z$ on the line:
>
> * If $X < Y < Z$, both constraints $(X, Y^+, Z^-)$ and $(Y, Z^+, X^-)$ are satisfied if and only if $|X-Y| \geq |Y-Z|$.
>
> * If $X < Z < Y$, $(X, Y^+, Z^-)$ cannot be satisfied.
>
> * If $Y < X < Z$, $(Y, Z^+, X^-)$ cannot be satisfied.
>
> * If $Y < Z < X$, $(X, Y^+, Z^-)$ cannot be satisfied.
>
> * If $Z < X < Y$, $(Y, Z^+, X^-)$ cannot be satisfied.
>
> * If $Z < Y < X$, both constraints are satisfied if and only if $|X-Y| \geq |Y-Z|$.
>
> > "In lines 223--228, the proof shows that violating a boundary constraint results in a net loss, but it assumes that the total loss is upper bounded by $\sum w_{uv}$. Why does this upper bound always dominate the gain from violating any single boundary constraint?"
>
> This property holds because we specifically set $M := \sum_{(u,v) \in E} w_{uv} + 1$ in our construction (lines 203--204). We would like to justify this argument as follows:
>
> Consider the consequences of moving $v$ in this way. (1) The boundary constraints $(X, Y^+, v^-)$ and $(X, v^+, Z^-)$ become satisfied, yielding gain $M$ (if one was previously unsatisfied) or $2M$ (if both were previously unsatisfied). (2) Edge constraints involving $v$, which in the worst case all become violated, yielding maximum loss $\sum_{(u,v) \in E} w_{uv}$.
>
> Therefore,
> $
>     \text{gain} \geq M, \text{loss} \leq \sum_{(u,v) \in E} w_{uv} \Rightarrow \text{gain} - \text{loss} \geq M - \sum_{(u,v) \in E} w_{uv} = 1 > 0.
> $
> This inequality shows that fixing a boundary constraint always improve the solution, thus any local optimum must satisfy all boundary constraints. We will make the inequality explicit in the revised version.
>
> > "Theorem 4.1, it would be very helpful if you could list the 12 triplets used to encode a quadratic over $x$, $y$, $z$, and share a simple example (either symbolic or numeric) of the corresponding weights. This would really clarify how cross-terms like $xy$ are handled in the reduction."
>
> We listed the 12 triplets in the Appendix section (line 645-656) due to the space constraint of the main paper. In our reduction, two triplets $(t_1 = (x, 0, y)$  and $t_2 = (y, 0, x))$ contribute to the cross-term $xy.$ We will add detailed illustrations regarding the triplet constructions to the main paper in the revised version if space allows.
>
> We would like to thank the reviewer again for their valuable time and supportive comments.
>
> Best regards,
>
> The authors.
> _____________
> References:
>
> [1] Pavel Huáček and Eylon Yogev. “Hardness of Continuous Local Search: Query Complexity and Cryptographic Lower Bounds”. In: Proceedings of the 2017 Annual ACM-SIAM Symposium on Discrete Algorithms (SODA), pp. 1352–1371.

---

### Note · Authors · 2025-08-15

We thank the reviewers and Area Chair for their time and feedback.

**On Scope and Track Suitability**

We note that some reviewers expected empirical or application-oriented results. We respectfully emphasize that our submission was to the Theory track, and the foundational nature of our results aligns with NeurIPS's tradition of publishing influential theoretical works that can inspire future developments in both theory and practice. As discussed in our responses, the worst-case hardness results obtained here are crucial precursors to average-case or smoothed analysis and can inform the design of algorithms and heuristics in the field.

**Summary of Discussion**

- **Technical Review:** We appreciate the reviewers' recognition of our work’s novelty and rigor. We hope our responses have clarified the requested proof details and improved explanations as suggested.

- **Practicality Concerns:** Some reviewers expressed concerns regarding practical relevance and empirical validation of our work. In our response, we clarified that our focus is on foundational theory, but we also provided additional experiments on perturbed instances to offer an empirical perspective. Furthermore, we explained how our worst-case analysis sets the stage for future work on average-case and practical scenarios. Finally, we showed that our reduction establishes a connection between the complexity of contrastive learning and the degree of the dependency hypergraph, highlighting necessary conditions on the triplets that render contrastive learning computationally hard.

**Strengths of Our Work**

- First formal complexity analysis of local optima in contrastive learning, resolving an open theoretical question.
- Technically novel and involved reductions for both discrete and continuous settings.
- Foundation for future studies on optimization in machine learning.

We appreciate the reviewers' thoughtful feedback, which helped us clarify and strengthen our work. We believe our responses address the main concerns and hope the committee recognizes the theoretical significance of our contribution.

Best regards,

The authors.

---

### Decision · Program_Chairs · 2025-09-17

**Decision:**

Accept (poster)

**Comment:**

The paper shows that in Contrastive Learning it is hard to fund even a local minima. This type of hardness result is stronger that the usual hardness of learning typically shown. Most of the reviewers therefore found the result of interest and I recommend acceptance.